# AdaEraser: Training-Free Object Removal via Adaptive Attention Suppression

**Dingming Liu** [1]

## Abstract

Object removal aims to eliminate specified objects from images while plausibly inpainting the affected regions with background content. Current training-free methods typically block attention to object regions within self-attention layers during the image generation process, leveraging surrounding background information to restore the image. However, indiscriminate suppression of self-attention in the vacated areas can degrade generation quality, as the model must simultaneously reconstruct background content in these regions. To solve this conflict, we propose AdaEraser, an adaptive framework that dynamically modulates attention based on the estimated presence of target object concepts. Through analysis of self-attention map evolution across denoising timesteps before and during removal, we develop a token-wise adaptive attention suppression strategy. This approach enables progressive perception of object removal throughout the denoising process, with the suppression strength in self-attention layers adjusted adaptively. Extensive experiments demonstrate that AdaEraser achieves superior performance in object removal, outperforming even training-based methods.

## 1. Introduction

Recent advances in diffusion models have revolutionized image generation and editing, offering unprecedented control over synthesis quality and semantic coherence (Ho et al., 2020; Rombach et al., 2022). These models iteratively denoise latent representations guided by text or spatial constraints, enabling applications such as inpainting, style transfer, and scene manipulation. Among these tasks, object removal presents a unique challenge: it requires not only the precise elimination of user-specified objects but also

the plausible reconstruction of background content in the vacated regions.

Object removal task shares certain common essence with image inpainting. Traditional inpainting methods rely on handcrafted algorithms, such as texture synthesis, *e.g.*, Patch-Match (Criminisi et al., 2004) and partial differential equations. While effective for small and repetitive textures, these methods falter in complex scenes due to limited semantic understanding. GAN-based models (Yu et al., 2018) and diffusion-based methods (Rombach et al., 2022), either trained from scratch or tuned from a pretrained generative model, improve texture generation but are limited by high training costs and dataset dependency. Training-free image inpainting has emerged as a data-efficient alternative. Techniques like Deep Generative Prior (Pan et al., 2021), RePaint (Lugmayr et al., 2022), MagicRemover (Yang et al., 2023), MagicEraser (Li et al., 2024a), and AttentiveEraser (Sun et al., 2025) devise varying ways to leverage generative priors from pre-trained generative models and remove selected contents in a training-free manner. Recent research trends (Li et al., 2024a; Sun et al., 2025) focus on modulating self-attention layers to suppress intra-region attention within object areas, thereby effectively facilitating object removal, as shown in Figure 1 (a). Though showing impressive removal results, especially AttentiveEraser (Sun et al., 2025), they struggle in generating high-quality background, as shown in Figure 1 (c).

Essentially, object removal can be decomposed into two goals, *i.e.*, removing the target object and generating the background. Indiscriminately suppressing self-attention in object regions, while capable of completely removing the target object, inadvertently disrupts the background generation process. This occurs because background generation inherently relies on the generative capabilities of the pretrained diffusion model, which was originally designed with unmodulated self-attention mechanisms. To preserve background generation capability, we believe that the key lies in monitoring the presence of the target object during removal and adaptively regulating the suppression strength of self-attention based on its presence level. For instance, when we observe that the object has been largely removed, we reduce the suppression intensity, thereby allowing the background to be generated in the normal way.

---

[1]School of Computer Science, Peking University. Correspondence to: Dingming Liu <dmliu25@stu.pku.edu.cn>.

*Proceedings of the 43rd International Conference on Machine Learning*, Seoul, South Korea. PMLR 306, 2026. Copyright 2026 by the author(s).

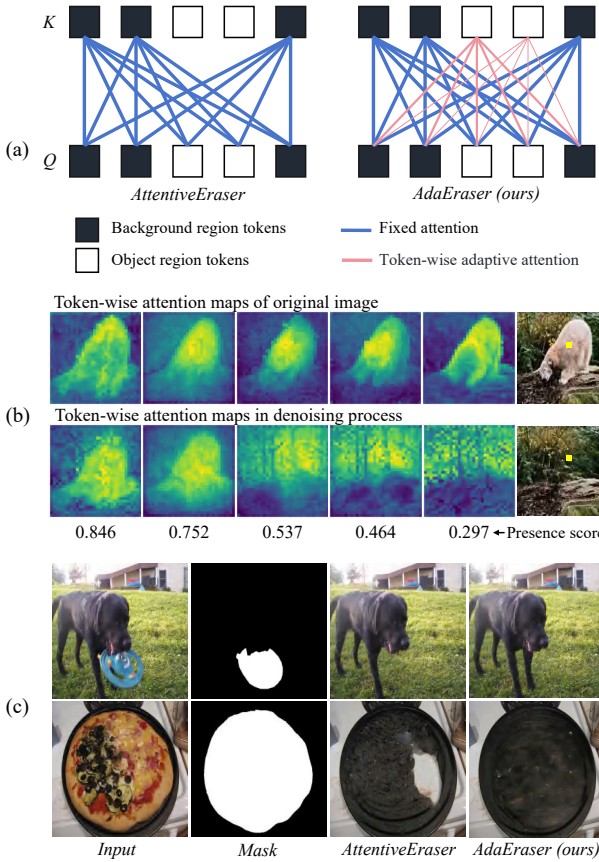

*Figure 1.* (a) AttentiveEraser (Sun et al., 2025) suppresses self-attentions from image tokens in $Q$ to object tokens in $K$, while our AdaEraser adaptively adjust the suppression strength based on the presence level of the target object. (b) We identify token-wise self-attention maps as an effective representation to approximate the presence score. (c) AttentiveEraser tends to introduce structural artifacts and visual distortions due to inadequate attention to the masked region during image synthesis. In contrast, AdaEraser shows superior quality even in extreme cases.

However, quantifying the presence of a specific object concept during the denoising process is non-trivial. First, the object cannot be directly detected in the latent space. Second, the token features from the original image (before removal) and those from the denoising process are not directly comparable, as they may reside in different feature spaces. Nevertheless, we observe that the self-attention maps of tokens within the object region to all other tokens encode the token's contextual relationships and implicitly capture semantic features of the image content, as shown in Figure 1 (b). Crucially, these maps are normalized via Softmax, rendering them comparable between those from the original image and those from the denoising process. We observe that simple cosine similarity between token-wise self-attention maps from the original image and from the denoising process has high correlation with the presence level of the object across denoising steps and layers.

Motivated by this observation, we propose AdaEraser, an effective training-free object removal framework with an adaptive attention suppression strategy. Specifically, our AdaEraser consists of three main steps:

1. First, we compare the self-attention maps within the object region before and during object removal to monitor the presence level of the target object dynamically;

2. Second, token-wise presence scores are computed for all spatial tokens within the object region;

3. Finally, we compute suppression coefficients based on the presence scores, then modulate the self-attention layers with these coefficients.

In this way, we conduct extensive experiments on varying benchmarks (Tudosiu et al., 2024; Zhao et al., 2024). Compared with previous state-of-the-art inpainting and object removal methods (Podell et al., 2023; Zhuang et al., 2024; Ekin et al., 2024; Li et al., 2025; Sun et al., 2025; Jiang et al., 2025; Yu et al., 2025), AdaEraser achieves consistently better performance in object removal and visual quality preservation, even outperforming training-based approaches.

The key contributions of our work are summarized as follows:

• We identify adaptive regulation based on the residual object presence during denoising as a key for high-quality object removal, and show that token-wise self-attention similarity can serve as an effective heuristic proxy for this purpose.

• We propose AdaEraser, a training-free object removal approach with an adaptive attention suppression strategy.

• AdaEraser achieves superior performance over training-free or training-based methods, providing a new perspective on object removal.

**Conflict of Interest Disclosure.** We declare no financial conflicts of interest related to this work.

## 2. Related Work

### 2.1. Object Removal

Object removal refers to the process of eliminating designated objects from images, and subsequently synthesizing plausible content to fill the resulting voids, ensuring visual coherence with the remaining image regions. Early works employed CNN-based (Suvorov et al., 2022) or transformer-based (Dong et al., 2022; Shamsolmoali et al., 2023; Ko

& Kim, 2023) image translation networks and utilized adversarial loss from GANs (Ma et al., 2022) to enhance the consistency of inpainted regions.

In recent years, diffusion-based methods have become mainstream due to their superior ability to capture complex data distributions and their more stable performance, and can be categorized into two categories:

1) training-based methods, most of which leverage pre-trained models and design adapters or finetune models to achieve better performance. PowerPaint (Zhuang et al., 2024) introduces learnable task prompts specifically designed for object removal tasks. ClipAway (Ekin et al., 2024) leverages AlphaCLIP (Sun et al., 2024) and IP-Adapter (Ye et al., 2023) to inject optimized background information, thereby improving the quality of object removal results. RORem (Li et al., 2025) and SmartEraser (Jiang et al., 2025) construct dedicated datasets for the object removal task and utilize them to enhance model performance. Other methods (Zhang et al., 2024; Sheynin et al., 2024; Brooks et al., 2023; Yildirim et al., 2023) can incorporate learnable embeddings or textual prompts as additional guidance to assist in the object removal task. OmniPaint (Yu et al., 2025) introduces a unified framework that re-conceptualizes object removal and insertion as interdependent processes. However, these methods demand substantial amounts of high-quality data, the acquisition of which is often prohibitively costly.

2) training-free methods (Avrahami et al., 2023; Li et al., 2024c; Jia et al., 2024; Han et al., 2024; Yang et al., 2023; Li et al., 2024b; Sun et al., 2025), aim to exploit the generative priors of pre-trained diffusion models for object removal. MagicRemover (Yang et al., 2023) achieves object removal by reducing the cross-attention map scores associated with the tokens of the objects to be removed. SuppressEOT (Li et al., 2024b) prevents the occurrence of undesired targets by manipulating text embeddings. Such text-driven methods are constrained by the information conveyed in the textual prompt, and tend to perform sub-optimally when the target object is small or when only a subset of multiple similar objects needs to be removed. MagicEraser (Li et al., 2024a) modulates self-attention layers based on additional semantic cues to enhance controllability of the generation process. AttentiveEraser (Sun et al., 2025) removes objects by blocking the attention from the entire image to the region containing the object to be removed in the self-attention layers. However, the background restoration in this region suffers from low visual quality due to the lack of self-attention within the region itself.

## 2.2. Attention Mechanism within Diffusion Models

Attention module (Vaswani et al., 2017) serves as an essential part of diffusion models. It updates the interme-diate features based on the context features and can be formulated as $\text{Attention}(Q, K, V) = \text{Softmax}\left(\frac{QK^T}{\sqrt{d}}\right)V$, where $Q$, $K$ and $V$ are the query, key and value features projected from the latent features, and $d$ denotes the dimension of the query features. The attention map $A$ is defined as $A = \text{Softmax}\left(\frac{QK^T}{\sqrt{d}}\right)$, which is used to aggregate the value $V$. $Q$ is derived from the latent code in the denoising process, while $K$ and $V$ are obtained from the latent code and the text encoding in the self-attention and cross-attention layers, respectively. By aggregating visual features from various regions of the image, the self-attention mechanism helps maintain global consistency and visual harmony in the generated outputs.

Self-attention maps control the spatial arrangement of image elements (Tewel et al., 2023; Tumanyan et al., 2023; Nam et al., 2024), and tend to resemble the image layout as the generation proceeds. Query-key pairs belonging to the same object tend to have larger scores during the generation process (Kim et al., 2023). Building upon this observation, we systematically analyze the variations in token-wise self-attention maps corresponding to the regions targeted for removal in object removal tasks. By utilizing these variations as a guiding mechanism, we facilitate a balanced trade-off between object removal and background reconstruction within these regions.

## 3. Method

The goal of object removal is to erase the specified target object in the masked region and seamlessly generate consistent content based on the surrounding environment. Formally, given a source image $I^{src}$ and a mask $M$ indicating the object to be removed, our objective is to generate a target image $I^{tgt}$ such that the object in the region indicated by $M$ in $I^{src}$ is removed and filled according to the background, while keeping the content in the remaining regions unchanged.

The key idea of AdaEraser is to design a token-level adaptive suppression mechanism within the self-attention layers, which adaptively suppresses attention to the region indicated by $M$, thus enabling both effective object removal and background restoration. We present our analysis of the self-attention maps in diffusion models in Sec. 3.1 and state our motivation in Sec. 3.2. Based on these insights, we then describe the detailed design of AdaEraser in Sec. 3.3.

### 3.1. Self-Attention Maps in Object Removal

**Token-wise self-attention maps**. Previous studies (Kim et al., 2023) have shown that query-key pairs belonging to the same object tend to exhibit higher attention scores during the generation process; that is, attention maps increasingly resemble the image layout as generation progresses. Further-

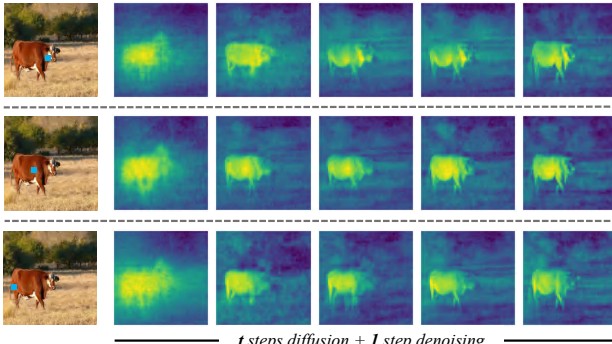

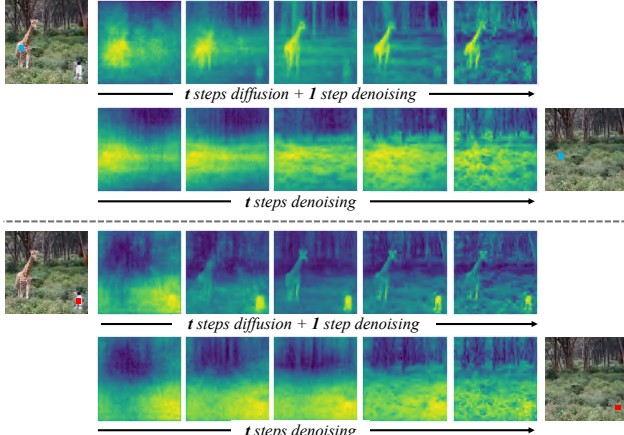

*Figure 2.* Visualization of token-wise self-attention maps. Given an image, we show the self-attention maps for the key tokens via feeding the image after $t$ steps diffusion into the denoising network. As the denoising timestep progresses, tokens affiliated with the same object exhibit increasingly stronger interactions, influencing the overall image layout. Moreover, even among tokens associated with the same object, the semantic focus can diverge — for instance, the self-attention maps corresponding to the cow's head, abdomen, and tail tokens highlight different regions.

*Figure 3.* Given a source image, we visualize the self-attention maps for the key tokens via $t$ steps diffusion and 1 step denoising, as well as the self-attention maps in the removal process after $t$ steps denoising. By varying $t$, we observe that during the removal process, the similarity between self-attention maps of a token located in the object region before and during removal gradually decrease, showing high correlation with decreasing presence of the object.

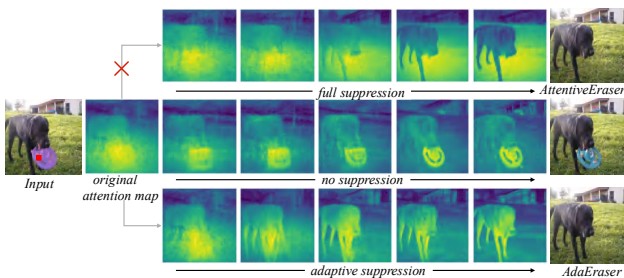

*Figure 4.* Visualization of different attention suppression strategies and corresponding generation outcomes.

more, diffusion models typically establish the position and overall appearance of each object in the early stages, while finer details—such as color and texture variations—are progressively refined in the later stages (Choi et al., 2022; Hertz et al., 2022).

We experimentally validate and further elaborate on this observation by demonstrating that, even among different tokens corresponding to the same object, there can be significant variation in their semantic focus within the attention mechanism. Specifically, our analysis reveals that query-key pairs which are associated with the same local structural region of an object consistently yield higher attention scores. This indicates that the attention mechanism is capable of capturing nuanced, token-level relationships that reflect the underlying spatial and semantic coherence of local object structures. The pattern of attention is not only present at the object level, but also persists at a more granular, token-wise scale, thereby highlighting the fine-grained discriminative power of the model's attention distributions. This behavior is further illustrated in Figure 2, which presents the self-attention maps of tokens corresponding to the cow's head, abdomen, and tail. Our analysis suggests that the generative prior of diffusion models is sufficient for each token to attend to the parts of the object that are more closely related to itself, rather than to the object as a whole. Therefore, we adopt token-wise operations in the subsequent experiments.

**Self-attention maps before and during object removal.** We further analyze the self-attention maps of tokens corresponding to the target object region before and during removal. In the early stages, the self-attention maps primarily reflect coarse contours, resulting in high similarity between the two cases. In the later stages, as noise is grad-

ually reduced, the self-attention maps increasingly capture the underlying semantic concept at that location. Consequently, as different contents occupy the same region before and after removal, the similarity between the corresponding self-attention maps decreases over time. Figure 3 illustrates an example of this phenomenon.

### 3.2. Motivation Clarification

We analyze the limitation of AttentiveEraser and motivate our design using the example shown in Figure 4. The original attention map, derived from the input image to be edited, evolve differently under different attention suppression strategies, leading to distinct generation outcomes.

Though the full-suppression strategy successfully removes the masked object, it over-relies on local background cues, and the lack of global context leads to incorrect background reconstruction. As illustrated in the first row of Figure 4, under AttentiveEraser, the key token gradually assigns higher

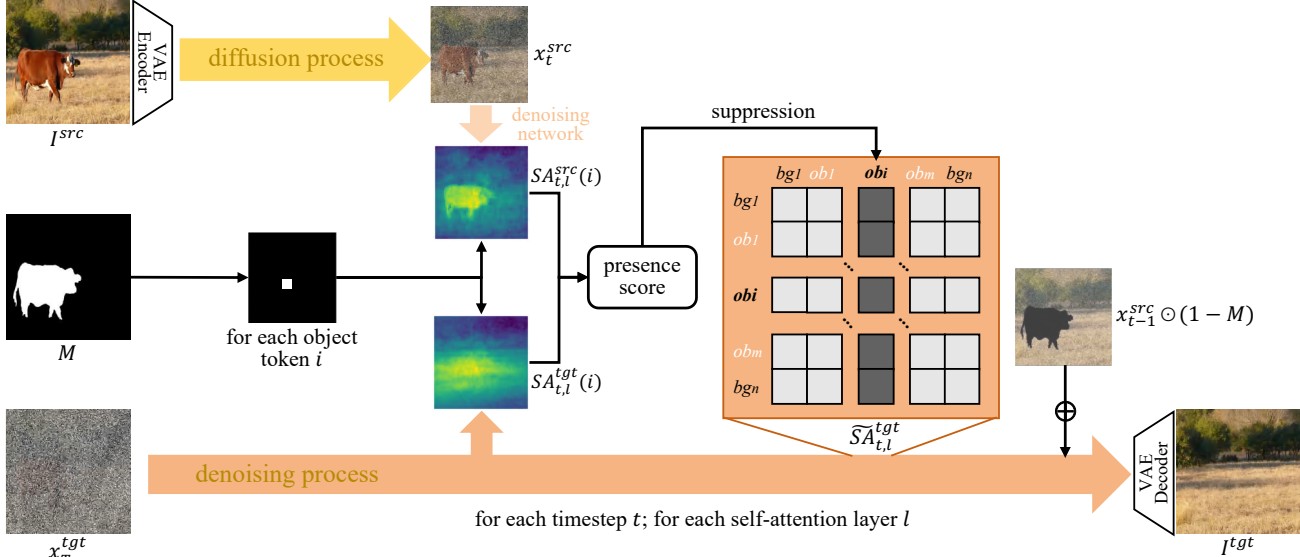

*Figure 5.* **Framework of AdaEraser**. Given a source image $I^{src}$ followed by the VAE-Encoder, for each time step $t = T, T-1, ..., 1$, (1) the diffusion process applies $t$ steps diffusion to the input and feed it to the denoising network. (2) The denoising process feeds $x_t^{tgt}$ to the denoising network, initialized with $x_T^{tgt}$, *i.e.*, $I^{src}$ after $T$ steps diffusion. (3) For each self-attention layer and object token, we extract the self-attention maps from both branches, and compute the presence score via comparing them. (4) The presence score is then used to suppress the self-attention scores corresponding to $i$-th key token. (5) We preserve the background by combining it with the denoised output using the mask. This process repeats until $t = 1$, followed by the VAE-Decoder to obtain $I^{tgt}$. Note that the denoising for $x_t^{src}$ and $x_t^{tgt}$ are carried out simultaneously by concatenating them and feed them into the denoising network.

attention scores to background grass tokens, ultimately restoring the masked area as grass rather than the semantically correct dog's leg.

The key challenge is to retain a suppression mechanism that effectively removes the object while enabling global context recovery. We aim to design a gradual-transition scheme that monitors the presence of the target object during removal and adaptively regulates the suppression strength of self-attention based on its presence level. However, directly measuring the object's presence in the latent space is challenging.

Instead of using an absolute metric, motivated by the analysis in Sec. 3.1, we adopt an alternative relative metric for detection. As illustrated in the second row of Figure 4, if no attention suppression is applied during denoising, the illustrative token's self-attention map still reflects features of the target object. Using this as an intermediate reference, we quantify object presence by comparing the differences between attention maps throughout the denoising process.

### 3.3. AdaEraser

We adopt a pretrained text-to-image diffusion model, including its VAE-Encoder, the denoising network, and the VAE-Decoder. The framework of AdaEraser is shown in Figure 5.

**Reference Attention Maps**. Given a source image $I^{src}$, we first encode it using the VAE-Encoder to obtain $x_0^{src}$:

$$x_0^{src} = \text{VAE-Encoder}\left(I^{src}\right) \quad (1)$$

To obtain the self-attention maps of $I^{src}$ as the reference, instead of performing a full inversion process - which typically involves searching for optimal latent noise variables - we adopt a simple "$t$ steps diffusion and 1 step denoising" scheme, where $t = T, T-1, ..., 1$. To facilitate this, we introduce Gaussian noise at multiple scales to the source image, where each noise scale corresponds to a specific timestep in the forward diffusion process. It can be formulated as:

$$x_t^{src} = \sqrt{\bar{\alpha}_t}\, x_0^{src} + \sqrt{1 - \bar{\alpha}_t}\, \epsilon, \quad (2)$$

where $\epsilon \sim \mathcal{N}(\mathbf{0}, \mathbf{I})$. This operation better preserves the feature information of the original image under noisy conditions, as it avoids cumulative errors (Avrahami et al., 2024; Tewel et al., 2024). The resulting noisy latent code $x_t^{src}$ at each timestep are subsequently fed into the denoising network to obtain self-attention maps from each attention layer $l$, denoted as $SA_{t,l}^{src}$ which serve as informative descriptors of object existence and localization.

**Target Attention Maps**. To preserve visual details from the source image, we design the starting point for denoising process, $x_T^{tgt}$, as:

$$x_T^{tgt} = \sqrt{\bar{\alpha}_T}\, x_0^{src} + \sqrt{1 - \bar{\alpha}_T}\, \epsilon, \quad (3)$$

where $\epsilon \sim \mathcal{N}(\mathbf{0}, \mathbf{I})$, $T$ is the total denoising steps. For each step $t$, $x_t^{tgt}$, the denoising output from the last step, is fed into the denoising network. We extract self-attention maps from each attention layer $l$, denoted as $SA_{t,l}^{tgt}$, as the representation to approximate the presence level of the target object.

**Token-Wise Presence Score**. $SA_{t,l}^{src}$ and $SA_{t,l}^{tgt}$ serve as representations in a unified space that reflects visual contents in latent features. We compare these representations to obtain a token-wise signal that correlates with the residual presence of the object to be removed. Importantly, this signal is not intended to be a rigorously calibrated estimator of true object presence, since object existence is not directly observable in the noisy latent space. Instead, we use it as a heuristic, control-oriented proxy that provides a stable relative signal for adaptively regulating self-attention suppression.

Additionally, as analyzed in Section 3.1, different tokens within the same object focus on different semantic regions. Therefore, we introduce a "token-wise presence score", formulated as:

$$p(i) = Sim\left(SA_{t,l}^{tgt}(i), \ SA_{t,l}^{src}(i)\right), \qquad (4)$$

where $SA_{t,l}^{tgt}(i)$ and $SA_{t,l}^{src}(i)$ denote the self-attention maps corresponding to the $i$-th token within the object mask $M$. $Sim$ is a similarity metric. In our experiments, we flatten the attention maps and adopt cosine similarity, as previous studies have shown that the cosine correspondence of latent features is sufficient to reflect the correspondence between real images (Tang et al., 2023).

**Token-Wise Adaptive Self-attention Suppression**. Provided the presence score, we calculate a suppression coefficient $\eta(i)$ as:

$$\eta(i) = \begin{cases} 1 - p(i), & \text{if } i\text{-th token } \in M \\ 1, & \text{otherwise} \end{cases} \qquad (5)$$

Subsequently, we implement the proposed adaptive suppression strategy by modifying the softmax operation of the self-attention map $SA_{t,l}^{tgt}$. In the self-attention layer of the denoising UNet, the modulated attention score $\widetilde{SA}_{t,l}^{tgt}(i)$ from the query tokens to the $i$-th key token is computed as follows:

$$\widetilde{SA}_{t,l}^{tgt}(i) = \frac{\eta(i)\exp\left(QK_i^\top/\sqrt{d}\right)}{\sum_{j=1}^N \eta(j)\exp\left(QK_j^\top/\sqrt{d}\right)}, \qquad (6)$$

where $Q$ denotes the query tokens and $K_i$ represent the $i-th$ key tokens, $N$ is the total number of image tokens, $d$ is the feature dimension. The modulated attention layer is

used in the $l$-th layer to produce the output for the next layer. Appendix A provides an interpretive theoretical analysis of AdaEraser, explaining why the proposed presence score can serve as a stable monotonic proxy for adaptive suppression and how the suppression rule can be viewed as a principled attention reweighting mechanism.

**Foreground-Background Blending**. By incorporating the aforementioned suppression strategy into the self-attention layers of the denoising UNet, we input $x_t^{tgt}$ and obtain $x_{t-1}^{tgt}$ through denoising at each timestep $t$. To maintain consistency in the background region, following (Avrahami et al., 2022; Jia et al., 2025; Sun et al., 2025), we combine the foreground of $x_{t-1}^{tgt}$ and the background of $x_{t-1}^{src}$ according to the object mask $M$, resulting in a new $x_{t-1}^{tgt}$. This process can be formulated as:

$$x_{t-1}^{tgt} = x_{t-1}^{tgt} \odot M + x_{t-1}^{src} \odot (1 - M) \qquad (7)$$

By iterating $t = T, T-1, ..., 1$, we finally obtain $x_0^{tgt}$, followed by the VAE-Decoder to produce the final edited result $I^{tgt}$:

$$I^{tgt} = \text{VAE-Decoder}\left(x_0^{tgt}\right) \qquad (8)$$

## 4. Experiments

### 4.1. Datasets and Evaluation Protocols

We use Stable Diffusion XL (SDXL) (Podell et al., 2023) as the base model for its balance of quality and efficiency, though our method is compatible with varying diffusion-based image generation network. Results based on other diffusion model architectures are included in the appendix. In the diffusion process, we provide an empty prompt as text condition.

AdaEraser is not tied to a specific diffusion architecture, as it only operates on token-wise self-attention maps and requires no additional training. Although we use SDXL as the main backbone for its balance of quality and efficiency, we further evaluate AdaEraser on SD1.5, SD2.1, FLUX, and SDXL. The results show consistently strong performance across different backbones, and the detailed quantitative comparison is provided in Appendix E.

**Baselines.** We compare AdaEraser with the following methods, which represent the previous state-of-the-art performance on both image inpainting and object removal, including SDXL-inpainting (SDXL-INP) (Podell et al., 2023), PowerPaint (Zhuang et al., 2024), CLIP-Away (Ekin et al., 2024), AttentiveEraser (Sun et al., 2025), SmartEraser (Jiang et al., 2025), RORem (Li et al., 2025) and OmniPaint (Yu et al., 2025). All methods employ the pre-trained diffusion model.

**Evaluation data.** We use the evaluation dataset as below:

*Table 1.* The quantitative comparison of different object removal methods on Mulan and OABench. AHR represents Average Human Ranking, *i.e.*, the average rank of users on a scale from 1 to 8 (lower is worse).

| Method | Training | Mulan | | | | | | OABench | | | | | |
|---|---|---|---|---|---|---|---|---|---|---|---|---|---|
| | | FID↓ | LPIPS↓ | PSNR↑ | ReMOVE↑ | CFD↓ | AHR↑ | FID↓ | LPIPS↓ | PSNR↑ | ReMOVE↑ | CFD↓ | AHR↑ |
| SDXL-INP (Podell et al., 2023) | ✓ | 115.85 | 0.2834 | 19.4398 | 0.6385 | 0.5011 | 1.98 ± 0.31 | 41.618 | 0.1597 | 23.2268 | 0.7022 | 0.4316 | 2.45 ± 0.36 |
| PowerPaint (Zhuang et al., 2024) | ✓ | 134.80 | 0.3147 | 18.0304 | 0.6069 | 0.4262 | 2.64 ± 0.42 | 51.265 | 0.1728 | 21.3932 | 0.6385 | 0.3707 | 2.94 ± 0.53 |
| Clipaway (Ekin et al., 2024) | ✓ | 62.694 | 0.2313 | 21.0102 | 0.8697 | 0.2759 | 3.87 ± 0.35 | 38.958 | 0.1595 | 22.8391 | 0.7783 | 0.3202 | 3.74 ± 0.45 |
| AttentiveEraser (Sun et al., 2025) | ✗ | 54.040 | 0.2221 | 22.7771 | 0.9000 | 0.2095 | 5.46 ± 0.27 | 40.373 | 0.1681 | 23.2670 | 0.8215 | 0.2604 | 5.43 ± 0.32 |
| SmartEraser (Jiang et al., 2025) | ✓ | 76.499 | 0.2498 | 20.2935 | 0.8542 | 0.3728 | 3.67 ± 0.56 | 38.659 | 0.1591 | 23.4476 | 0.7723 | 0.3471 | 3.79 ± 0.40 |
| RORem (Li et al., 2025) | ✓ | 53.470 | 0.2086 | 23.5275 | 0.9048 | 0.2033 | 6.22 ± 0.19 | 39.215 | 0.1569 | 23.4126 | 0.8281 | 0.2505 | 6.23 ± 0.28 |
| OmniPaint (Yu et al., 2025) | ✓ | 59.996 | 0.2291 | 21.4493 | 0.8706 | 0.2329 | 5.07 ± 0.38 | 38.903 | 0.1593 | 22.9257 | 0.7991 | 0.2618 | 4.59 ± 0.47 |
| AdaEraser (Ours) | ✗ | **51.108** | **0.2026** | **23.5871** | **0.9065** | **0.2002** | **7.08 ± 0.34** | **38.472** | **0.1562** | **23.5047** | **0.8316** | **0.2450** | **6.81 ± 0.21** |

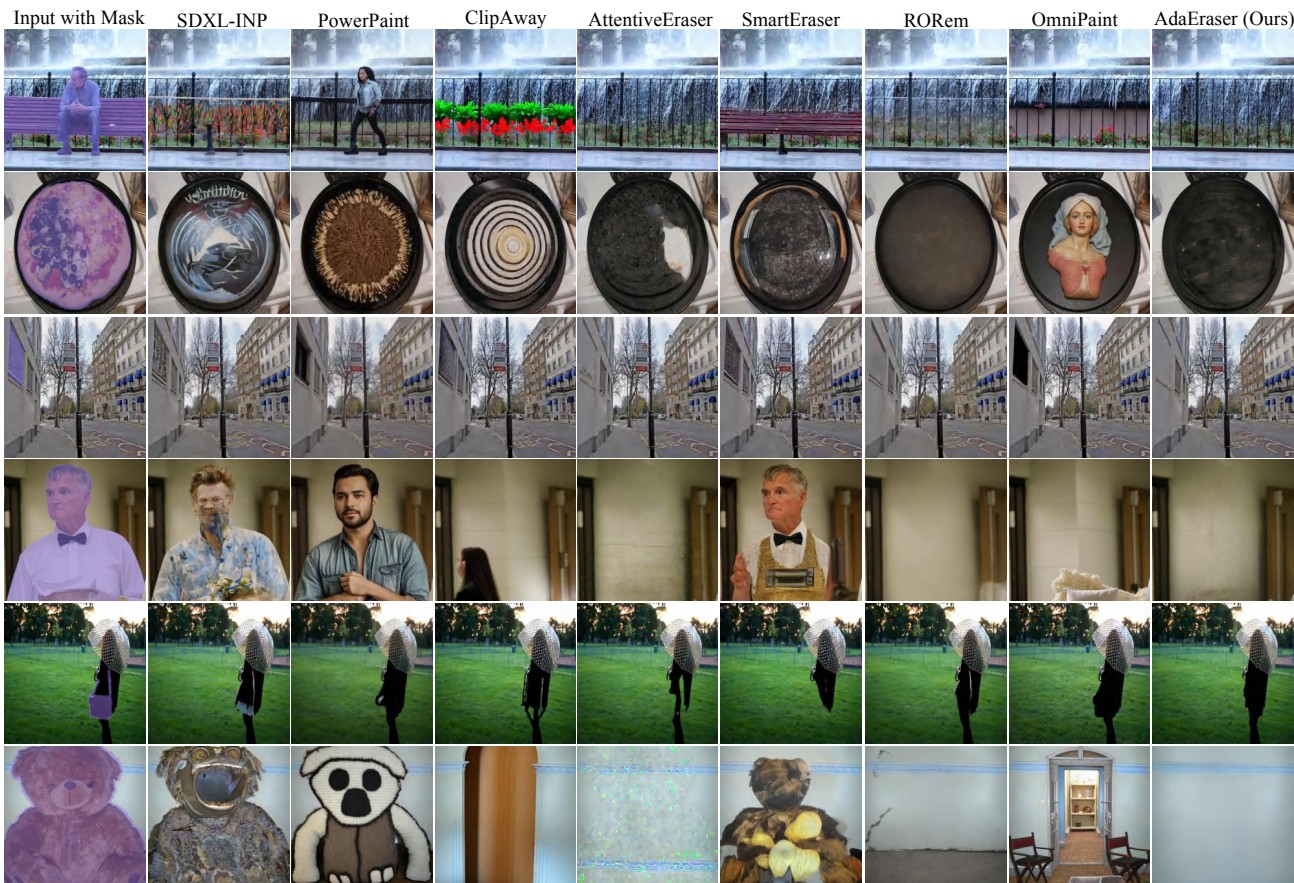

*Figure 6.* Qualitative comparison. It can be observed that previous methods often yield incomplete or excessive removal regions as well as undesired synthesized contents. In contrast, AdaEraser employs an adaptive suppression strategy to effectively balance object removal and background restoration, resulting in significant improvements in both visual fidelity and overall performance.

1. Mulan dataset. Mulan (Tudosiu et al., 2024) is a dataset with images are from COCO (Lin et al., 2014) and Laion Aesthetics V6.5 (Schuhmann et al., 2022). Original images, extracted foreground object masks, and the groundtruth images with the foreground object removed are collected.

2. OABench dataset. OABench (Zhao et al., 2024) is a high-quality dataset composed of original images, inpainted images, specified objects to be removed, and the corresponding object masks in real-world scenarios.

**Evaluation metrics.** For quantitative evaluations of different object removal models, we consider two key aspects. First, we utilize FID (Heusel et al., 2017), LPIPS (Zhang et al., 2018), and PSNR (Hore & Ziou, 2010) to evaluate the consistency between the predicted image and the groundtruth. Second, we adopt the ReMOVE (Chandrasekar et al., 2024) and CFD (Yu et al., 2025) for the coherence between the generated regions and the background.

We conducted user studies as a more comprehensive evaluation of the object removal performance. We randomly sample 10 comparison cases from each of the two dataset (Tu-

dosiu et al., 2024; Zhao et al., 2024) and invite 30 users to rank the results. Following ControlNet (Zhang et al., 2023), we report the Average Human Ranking (AHR) in Table 1.

## 4.2. Comparisons with State-of-the-art Methods

**Quantitative comparisons.** The results are presented in Table 1. On Mulan (Tudosiu et al., 2024) and OABench (Zhao et al., 2024), AdaEraser outperforms all other training-based and training-free competitors (Podell et al., 2023; Zhuang et al., 2024; Ekin et al., 2024; Li et al., 2025; Sun et al., 2025; Jiang et al., 2025; Yu et al., 2025) across all metrics, which indicates the effectiveness of AdaEraser for the object removal task. Compared with all existing object removal approaches built upon pretrained diffusion models, AdaEraser exploits the potential of pretrained priors to the fullest extent in a training-free manner, achieving an optimal balance between object removal and background restoration.

**Qualitative comparisons.** Visual comparisons are showcased in Figure 6. All baseline methods exhibit certain limitations. Specifically, SDXL-INP (Podell et al., 2023), PowerPaint (Zhuang et al., 2024), and Clipaway (Ekin et al., 2024) may introduce unintended objects into the edited images, compromising visual consistency. SmartEraser (Jiang et al., 2025) often fails to completely remove target objects and produces blurred artifacts. RORem (Li et al., 2025) performs suboptimally in preserving details and restoring content. OmniPaint (Yu et al., 2025) tends to fill target regions with content inconsistent or semantically irrelevant to the surrounding background. AttentiveEraser (Sun et al., 2025), due to fully suppressing self-attention layers, frequently results in structural distortions. In contrast, AdaEraser excels at removing target objects while preserving scene context.

## 4.3. Analysis

**Efficiency comparison.** Table 2 shows efficiency comparison with baselines, measured on the same evaluation set and averaged over multiple runs. In AdaEraser, denoising for the source and target latents is performed in parallel by concatenating them, with attention suppression applied only to the target. Compared to AttentiveEraser, AdaEraser adds one parallel computation for presence score estimation before suppression, resulting in minimal computational and memory overhead (within 15%).

*Table 2.* Efficiency comparison.

| Method | Inference Time(s) | Memory Overhead(MiB) |
|---|---|---|
| AttentiveEraser | 13.98 | 7966 |
| OmniPaint | 20.37 | 33478 |
| AdaEraser(ours) | 15.41 | 9014 |

**Self-attention suppression strategy.** To evaluate the effectiveness of our token-wise adaptive self-attention suppression strategy, we compare varied strategies and report

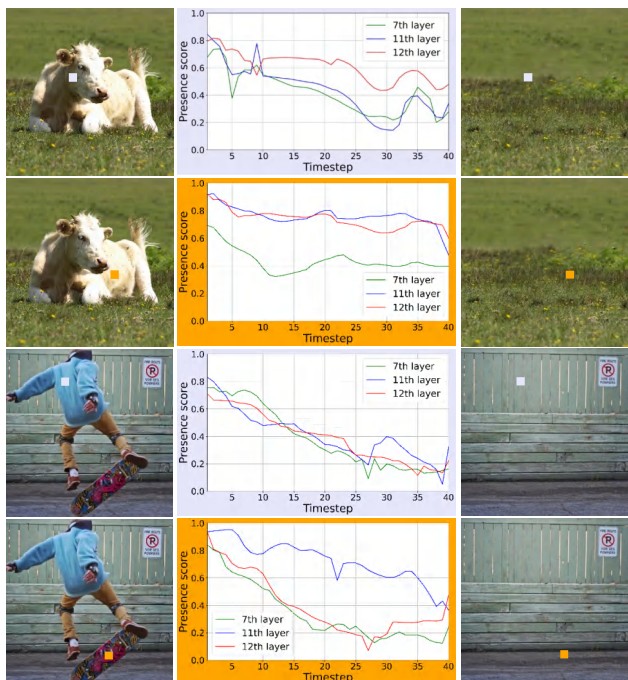

*Figure 7.* Evolution of presence score across timesteps. The curves represent different tokens in various layers.

both quantitative and qualitative results. Specifically, we compare the following two variants:

- **Timestep-based Suppression:** $p(i)$ decays from 1 to 0 linearly as diffusion timestep increases.

- **Region-based Suppression:** In this strategy, $p(i)$ is computed based on the overall masked region, rather than in a token-wise manner.

*Table 3.* The impact of different suppression strategies.

| Method | FID↓ | PSNR↑ | ReMOVE↑ | CFD↓ |
|---|---|---|---|---|
| Timestep-based suppression | 38.831 | 23.4697 | 0.8263 | 0.2517 |
| Region-based suppression | 38.945 | 23.4674 | 0.8261 | 0.2499 |
| Token-wise suppression | **38.472** | **23.5047** | **0.8316** | **0.2450** |

We use metrics described in Sec. 4.1 to quantify the performance of each strategy. Timestep-based suppression suffers from insufficient awareness of semantic concepts, while region-based suppression lacks fine-grained token-level perception. Both strategies yield inferior quantitative results. In comparison, the proposed token-wise suppression strategy achieves the best performance, benefiting from its fine-grained perception and suppression, which can be proved by Table 3. Please refer to Appendix B.1 for visualization.

**Reference selection.** To verify the effectiveness of selecting $x_t^{src}$ at timestep $t$ for obtaining reference, we conduct an ablation study with fixed-noise-level references: $x_1^{src}$, $x_T^{src}$

*Table 4.* The impact of different reference schemes.

| Reference | FID↓ | LPIPS↓ | PSNR↑ | ReMOVE↑ | CFD↓ |
|---|---|---|---|---|---|
| $x_1^{src}$ | 38.595 | 0.1578 | 23.4262 | 0.8223 | 0.2658 |
| $x_T^{src}$ | 38.829 | 0.1577 | 23.4808 | 0.8241 | 0.2507 |
| $x_{T/2}^{src}$ | 38.713 | 0.1574 | 23.4872 | 0.8262 | 0.2514 |
| $x_t^{src}$ | **38.472** | **0.1562** | **23.5047** | **0.8316** | **0.2450** |

and $x_{T/2}^{src}$ (the lowest, highest, and moderate noise level respectively). The results in Table 4 demonstrate that $x_t^{src}$ performs best, benefiting from precise noise-level alignment with the target. See Appendix B.2 for visualization.

**Evolution of presence score.** As shown in Figure 7, we visualize the presence score across timesteps. The scores exhibit a progressive decline correlating with the object removal progress. Each layer shows a unique mode of decline. Different tokens also show varying score curves. These results can further support the design of our adaptive suppression strategy.

### 4.4. Discussion

**Impact of mask quality on performance.** Since AdaEraser uses the user-provided mask to define the removal region, the quality and coverage of the mask directly affect the final editing result. We observe that different types of mask imprecision have different effects.

For slightly loose masks, where the mask extends beyond the target object boundary, AdaEraser remains robust. This is because the proposed adaptive attention suppression operates on the full-image attention manifold rather than hard-isolating the masked region, allowing background tokens inside the loose mask to still leverage contextual information from unmasked regions.

In contrast, incomplete masks are more challenging. When the mask fails to cover the entire target object or its induced side effects, such as shadows, reflections, or surrounding context changes, these residual regions may remain visible after editing and degrade removal quality. Therefore, AdaEraser is generally robust to accurate or slightly loose masks, but more sensitive to under-masked cases. A promising future direction is to automatically identify and include object-induced side effects in the removal mask, reducing the reliance on precise user annotations.

We provide additional analysis and visual examples in Appendix C.

**Performance under similar textures and failure cases.** Scenes with similar textures, repetitive background patterns, or overlapping same-class objects are challenging, since the target object and the surrounding content may share similar local appearance and thus make the token-wise presence score less discriminative. Nevertheless, AdaEraser still per-

forms well in many such cases, as shown in Appendix G, suggesting that adaptive suppression can preserve useful background cues while removing the target object.

However, since AdaEraser relies on the generative model's prior to restore the removed regions, it may still struggle when the content to be recovered is structurally complex or semantically ambiguous. In such cases, the model may produce semantic inaccuracies, structural distortions, or degraded visual quality, as illustrated in Figure 8. A promising future direction is to incorporate stronger structural guidance or semantic priors to better handle complex and ambiguous reconstruction scenarios.

Input with mask      AdaEraser

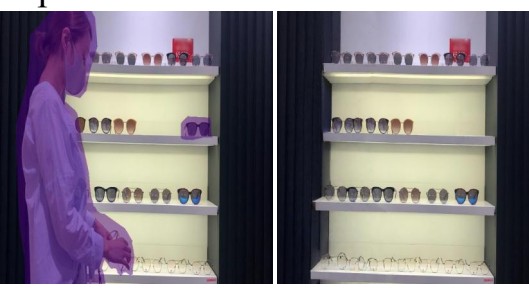
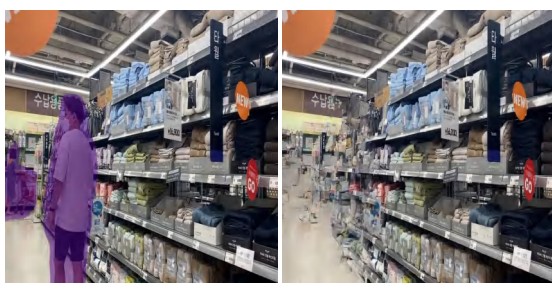

*Figure 8.* Failure cases of AdaEraser under complex and ambiguous reconstruction scenarios. When the content to be recovered contains intricate structures or semantically ambiguous regions, AdaEraser may produce structural distortions or degraded visual quality.

## 5. Conclusion

In this paper, we systematically investigate the dynamics of token-wise self-attention maps in pretrained diffusion models. Leveraging these findings, we introduce AdaEraser, a novel training-free object removal framework that adaptively balances object elimination and background restoration. By dynamically adjusting the suppression strength according to the estimated object presence, AdaEraser avoids the drawbacks of indiscriminate attention suppression and better preserves the generative capability of the pretrained model. Comprehensive experiments and user studies validate the superior effectiveness of our method on challenging object removal benchmarks. Our work provides a new perspective on object removal.

## Impact Statement

This paper presents work whose goal is to advance the field of object removal. There are many potential societal consequences of our work, none which we feel must be specifically highlighted here.

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

# Overview

This appendix consists of seven sections. §A provides a theoretical justification of AdaEraser, including the semantic interpretation of the presence score, robustness under diffusion noise, and the variational optimality of adaptive suppression. §B presents visual ablation studies analyzing different self-attention suppression strategies and reference selections. §C analyzes the effect of mask quality on AdaEraser, covering both incomplete masks that miss object-induced side effects and loose masks obtained by mask dilation. §D discusses the compatibility of AdaEraser with accelerated distilled diffusion models. §E shows the generalizability of AdaEraser across different diffusion architectures. §F details the user study. §G presents additional qualitative results demonstrating the stability and effectiveness of AdaEraser.

# A. Theoretical Justification of AdaEraser

In this section, we provide a theoretical interpretation of AdaEraser. Rather than claiming exact recovery of latent semantics, our goal is to explain why the proposed presence score and adaptive suppression mechanism form a stable and consistent semantic control strategy under realistic diffusion priors. We emphasize that these analyses are interpretive and explanatory in nature, rather than formal guarantees of convergence or optimality.

Throughout this appendix, $\langle \cdot, \cdot \rangle$ and $\| \cdot \|$ denote the standard Euclidean inner product and $\ell_2$ norm, respectively. All comparisons of presence scores are performed within the same attention layer and diffusion timestep.

## A.1. Presence Score as a Monotonic Semantic Indicator

The presence score $p(i)$ (Eq. 4) is designed to quantify the *relative semantic persistence* of the $i$-th token. We emphasize that $p(i)$ is not intended to be an unbiased estimator of the true semantic density, but a monotonic, order-preserving proxy sufficient for adaptive suppression. In practice, AdaEraser primarily relies on its relative ordering of $p(i)$ across tokens.

**Assumption 1 (Linear Mixture Model).** Following prior observations that Transformer attention encodes concepts in approximately additive subspaces, we model the target self-attention descriptor for token $i$ as

$$SA_{t,i}^{tgt} = \pi_{t,i} SA_{t,i}^{src} + (1 - \pi_{t,i}) SA_{t,i}^{bg}, \qquad \pi_{t,i} \in [0, 1], \tag{9}$$

where $\pi_{t,i}$ denotes the local semantic density of the object. This formulation is intended solely as a conceptual abstraction to capture the dominant semantic trend, rather than as an exact model of the underlying attention distribution. This mixture is defined in descriptor space after Softmax normalization and is used only to capture a first-order trend.

**Assumption 2 (Weak Cross-Semantic Correlation).** In high-dimensional descriptor space, attention descriptors associated with distinct semantics (object vs. background) tend to exhibit limited correlation:

$$\langle SA_{t,i}^{src}, SA_{t,i}^{bg} \rangle \leq \kappa \|SA_{t,i}^{src}\| \|SA_{t,i}^{bg}\|, \tag{10}$$

where $\kappa$ is moderately small.

**Assumption 3 (Comparable Descriptor Norms).** Within the same attention layer and resolution, descriptor norms vary mildly:

$$\|SA_{t,i}^{src}\| = L_s, \qquad \|SA_{t,i}^{bg}\| = L_b, \qquad \frac{L_s}{L_b} \in [1 - \epsilon_L, 1 + \epsilon_L], \quad \epsilon_L \ll 1. \tag{11}$$

**Monotonic Indicator Property.** Using Eq. 9, the inner product expands as

$$\langle SA_{t,i}^{tgt}, SA_{t,i}^{src} \rangle = \pi_{t,i} \|SA_{t,i}^{src}\|^2 + (1 - \pi_{t,i}) \langle SA_{t,i}^{bg}, SA_{t,i}^{src} \rangle$$
$$\approx \pi_{t,i} L_s^2 \quad (+ \mathcal{O}(\kappa L_s L_b)). \tag{12}$$

Similarly,

$$\|SA_{t,i}^{tgt}\|^2 = \pi_{t,i}^2 L_s^2 + (1 - \pi_{t,i})^2 L_b^2 + 2\pi_{t,i}(1 - \pi_{t,i}) \langle SA_{t,i}^{src}, SA_{t,i}^{bg} \rangle$$
$$\approx \pi_{t,i}^2 L_s^2 + (1 - \pi_{t,i})^2 L_b^2 \quad (+ \mathcal{O}(\kappa L_s L_b)). \tag{13}$$

Therefore,

$$p(i) = \cos(SA_{t,i}^{tgt}, SA_{t,i}^{src}) \approx \frac{\pi_{t,i}L_s}{\sqrt{\pi_{t,i}^2 L_s^2 + (1-\pi_{t,i})^2 L_b^2}} \quad (+\mathcal{O}(\kappa, \epsilon_L)). \tag{14}$$

**Discussion.**  For any $L_s, L_b > 0$, the mapping $g(\pi) = \frac{\pi L_s}{\sqrt{\pi^2 L_s^2 + (1-\pi)^2 L_b^2}}$ is strictly increasing on $[0,1]$. Indeed, direct differentiation yields

$$g'(\pi) = \frac{L_s L_b^2 (1-\pi)}{(\pi^2 L_s^2 + (1-\pi)^2 L_b^2)^{3/2}} \geq 0,$$

with strict positivity for all $\pi \in [0,1)$. Hence, $g$ is strictly increasing on $[0,1]$ and preserves the semantic ordering induced by $\pi_{t,i}$.

### A.2. Robustness under Diffusion Noise

We analyze the stability of the presence score under diffusion-induced noise.

**Noise Model.**  Let

$$SA^{src} = S^{src} + N^{src}, \qquad SA^{tgt} = S^{tgt} + N^{tgt}, \tag{15}$$

with

$$N^{src}, N^{tgt} \sim \mathcal{N}(0, \sigma_t^2 I), \qquad \mathbb{E}[N^{src}(N^{tgt})^\top] = \rho \sigma_t^2 I, \quad \rho \in [0,1].$$

Here $\rho$ measures the correlation between the noise-induced perturbations in the two branches. In our implementation, we reuse the same Gaussian noise realization $\epsilon$ when constructing the reference latents and initializing the target branch. This induces positively correlated perturbations between the compared descriptors and ensures that the attention maps are evaluated under the same noise level. Empirically, such noise-alignment reduces the variance of the cosine-based presence score and stabilizes token-wise rankings. Under the noise model above, this corresponds to a positively correlated regime with $\rho > 0$. The analysis below holds for the full range $\rho \in [0,1]$.

The reference branch is used solely to probe self-attention descriptors, rather than performing inversion or optimizing latents. We adopt an additive Gaussian perturbation as a tractable proxy for the aggregated stochasticity induced by diffusion noise and network nonlinearities.

**High-Dimensional Approximation.**  Define

$$X = \langle SA^{tgt}, SA^{src} \rangle, \quad Y = \|SA^{tgt}\|^2, \quad Z = \|SA^{src}\|^2,$$

where $D$ denotes the dimensionality of the attention descriptor. Their expectations satisfy

$$\mathbb{E}[X] = \langle S^{tgt}, S^{src} \rangle + \rho \sigma_t^2 D, \tag{16}$$

$$\mathbb{E}[Y] = \|S^{tgt}\|^2 + \sigma_t^2 D, \tag{17}$$

$$\mathbb{E}[Z] = \|S^{src}\|^2 + \sigma_t^2 D. \tag{18}$$

For large descriptor dimension $D$, both $Y$ and $Z$ concentrate sharply around their expectations due to standard concentration results for quadratic forms of Gaussian vectors. Similarly, the inner product term $X = \langle SA^{tgt}, SA^{src} \rangle$ also concentrates around its expectation. Specifically, under the Gaussian noise model, $\mathrm{Var}(X) = \mathcal{O}(D)$, yielding fluctuations of order $\mathcal{O}(\sqrt{D})$, while $Y$ and $Z$ scale as $\Theta(D)$. As a result, the relative contribution of fluctuations in $X$ to the ratio $X/\sqrt{YZ}$ vanishes as $D \to \infty$. Under this concentration regime, the nonlinear mapping $(X, Y, Z) \mapsto X/\sqrt{YZ}$ can be linearized around $(\mathbb{E}[X], \mathbb{E}[Y], \mathbb{E}[Z])$. A first-order Taylor approximation suggests

$$p(i) = \frac{X}{\sqrt{YZ}} \approx \frac{\mathbb{E}[X]}{\sqrt{\mathbb{E}[Y]\mathbb{E}[Z]}}. \tag{19}$$

The approximation holds with high probability as $D \to \infty$.

**Implications.**    The additive $\sigma_t^2 D$ terms act as a natural damping factor, preventing instability in high-noise regimes. Importantly, even when $\rho = 0$, the ordering of $p(i)$ remains stable provided the underlying semantic signal exhibits a finite margin.

### A.3. Variational Interpretation of Adaptive Attention Suppression

We interpret adaptive suppression as the solution to a KL-regularized optimization problem.

This formulation is intended as an interpretive model rather than a claim of global optimality for the full diffusion dynamics.

**Objective.**    For a fixed query token $i$, we solve

$$\min_{\widetilde{SA}_{i\cdot} \in \Delta} \text{KL}(\widetilde{SA}_{i\cdot} \,\|\, SA_{i\cdot}) + \beta \sum_j \widetilde{SA}_{ij} p(j), \tag{20}$$

where $\Delta$ is the probability simplex. Here $\beta$ is used only for interpretation and can be absorbed into the gating function.

**Solution and Equivalence.**    The optimal solution is

$$\widetilde{SA}_{ij} \propto SA_{ij} \exp(-\beta p(j)). \tag{21}$$

Since $SA_{ij} = \text{Softmax}(A_{ij})$, this is equivalent to a logit shift

$$A_{ij} \mapsto A_{ij} - \beta p(j).$$

Our implementation

$$\widetilde{SA}_{ij} = \frac{\eta(j) \exp(A_{ij})}{\sum_k \eta(k) \exp(A_{ik})}$$

corresponds to $\log \eta(j)$ as an additive logit bias, and is therefore mathematically equivalent.

**Linear Gating Approximation.**    We adopt $\eta(j) = 1 - p(j)$ as a simple, monotone, and numerically stable, parameter-free surrogate of $\exp(-\beta p(j))$, which preserves the same boundary behaviors on $p(j) \in [0, 1]$ and works well empirically. In particular, it can be viewed as a first-order monotone approximation in the low-to-moderate suppression regime.

### A.4. Bounded Residual Behavior under Iterative Suppression

Let $\pi_t = [\pi_{t,1}, \dots, \pi_{t,N}]$ denote an abstract token-wise semantic density, and define $V(\pi_t) = \frac{1}{2}\|\pi_t\|_2^2$. Since $\pi_t$ is not an explicit state variable of the diffusion model, we do not attempt to derive its exact dynamics. Instead, we use the standard uniformly ultimately bounded (UUB) template as an interpretive lens for the empirically observed plateauing behavior:

$$V(\pi_{t+1}) \le (1 - \gamma)V(\pi_t) + \zeta, \tag{22}$$

where $\gamma > 0$ and $\zeta$ summarizes unmodeled effects (approximation error and model bias). This template suggests that the induced semantic residue may converge to a bounded neighborhood rather than vanishing exactly, which is consistent with the fact that the cosine-based presence score often saturates to a small floor in late denoising steps. Such a non-zero floor indicates weak remnants that are insufficient to dominate generation, and motivates relaxing suppression to avoid over-suppression artifacts while preventing re-emergence.

## B. Visualization of Ablation Study

### B.1. Self-Attention Suppression Strategy

Figure 9 shows visual comparisons. Timestep-based suppression often yields suboptimal removal results due to its lack of semantic awareness regarding the concepts to be removed, which can lead to structural abnormalities and incomplete removal. Region-based suppression alleviates these issues to some extent, but still suffers from similar limitations due to the absence of fine-grained, token-level perception. In contrast, our token-level adaptive manner ensures precise perception, thereby enabling high-quality object removal.

## B.2. Reference Selection

Figure 10 shows visual comparisons using different references. Regardless of whether $x_1^{src}$, $x_T^{src}$, or $x_{T/2}^{src}$ is chosen as the reference with a fixed noise level throughout the denoising process, the removal performance remains unsatisfactory, leading to over-deletion, incomplete removal, insufficient detail restoration, and so on. In contrast, selecting $x_t^{src}$ with the same noise level as $x_t^{tgt}$ as the reference yields the best results, because maintaining the same noise level enables a more accurate and reasonable computation of the presence score.

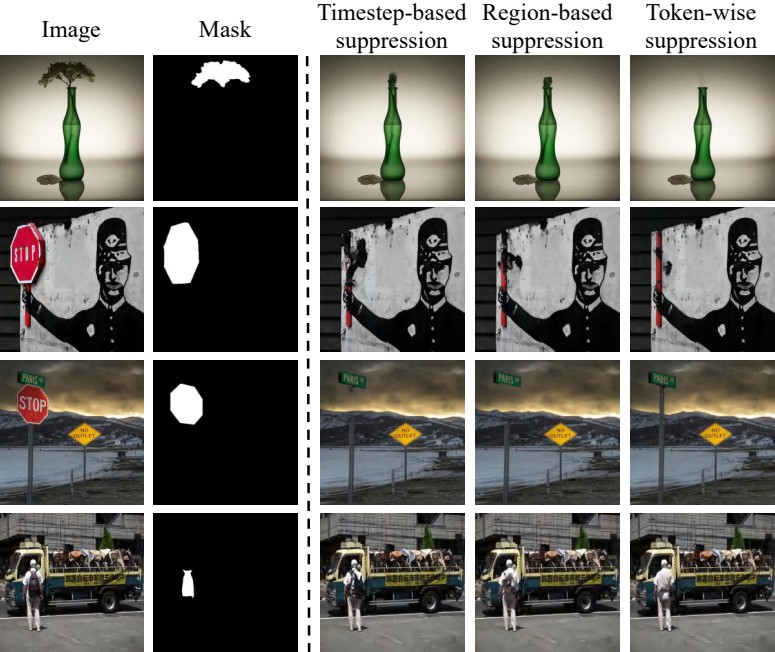

*Figure 9.* Visual comparisons of different self-attention suppression strategies.

## C. Effect of Mask Quality on AdaEraser

Since we adopt a training-free paradigm and a foreground–background blending strategy, AdaEraser can only erase the object regions explicitly specified by the input mask. When the mask does not include the side effects induced by the object (e.g., shadows, reflections), the removal quality may degrade. In contrast, when the mask covers both the object and its associated side effects, the editing results are much better, as shown in Figure 11.

To evaluate the robustness of AdaEraser to imprecise or loose masks, we apply morphological dilation to the object masks and conduct editing. Selected examples are shown in Figure 12, demonstrating that the method maintains stable removal quality even when the mask extends beyond the target object. Unlike hard-masking methods that physically isolate the masked region, AdaEraser operates on the full-image attention manifold. This allows the model to retain a global receptive field, ensuring that background tokens within a loose mask can still leverage long-range structural dependencies from outside the mask to maintain their original semantic integrity.

Overall, AdaEraser remains effective when the mask is accurate or slightly loose, but is more sensitive when the mask is incomplete.

## D. Compatibility with Accelerated Distilled Models

We also investigate the compatibility of AdaEraser with accelerated distilled diffusion models. Specifically, we evaluate AdaEraser on SDXL-Lightning with 4 denoising steps. Although this setting significantly reduces inference time, we observe a clear degradation in removal and reconstruction quality compared with the standard SDXL setting, with more noticeable artifacts appearing in the edited regions.

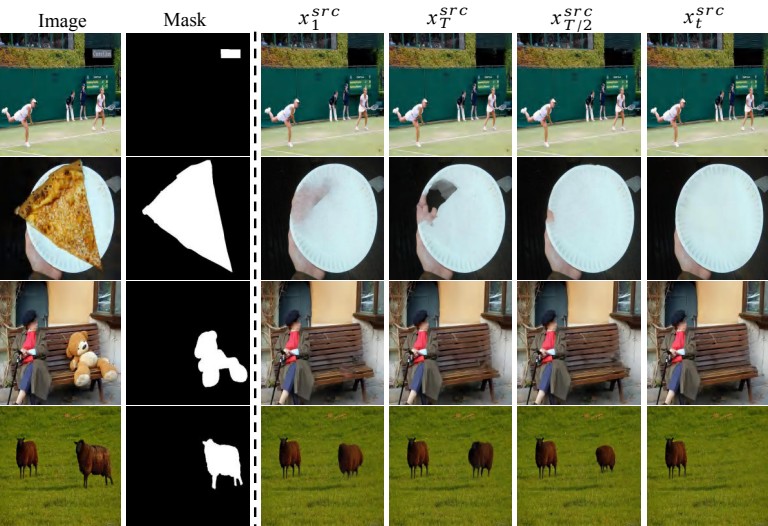

Image     Mask     $x_1^{src}$     $x_T^{src}$     $x_{T/2}^{src}$     $x_t^{src}$

*Figure 10.* Visual comparisons of different reference selections.

We believe this degradation is mainly because AdaEraser relies on sufficiently rich multi-step attention evolution during denoising. When the denoising process is compressed into very few steps, the attention dynamics become less informative, which weakens the effectiveness of our attention-adaptive erasing strategy. Therefore, AdaEraser does not directly transfer well to highly accelerated distilled models. Improving compatibility with such models is an important direction for future work.

## E. Generalizability to Other Architectures

To further evaluate the generalizability of AdaEraser, we apply it to different pretrained diffusion backbones, including SD1.5, SD2.1, FLUX, and SDXL. As shown in Table 5, AdaEraser achieves consistently strong performance across these architectures, suggesting that the proposed adaptive attention suppression strategy is not tied to a specific backbone. We use SDXL as the main model in our experiments due to its favorable balance between generation quality and efficiency.

*Table 5.* Cross-backbone generalization on the Mulan dataset.

| Architecture | FID↓ | LPIPS↓ | PSNR↑ | ReMOVE↑ | CFD↓ |
|---|---|---|---|---|---|
| SD1.5 | 55.981 | 0.2201 | 20.8360 | 0.9012 | 0.2057 |
| SD2.1 | 54.905 | 0.2132 | 22.5313 | 0.9019 | 0.2043 |
| FLUX | 52.335 | **0.2002** | 23.5690 | **0.9070** | 0.2008 |
| SDXL | **51.108** | 0.2026 | **23.5871** | 0.9065 | **0.2002** |

Figure 13 compares AdaEraser with SD1.5, SD2.1, SDXL and FLUX. It can be observed that our method achieves stable erasure across various architectures, showing promising generalization.

## F. Details about User Study

The user study questionnaire with 1/20 example is shown in Figure 14. To avoid ordering bias, the presentation order of the eight candidate results is randomly shuffled for each evaluation instance.

## G. More Results

Figure 15 demonstrates additional results that highlight the effectiveness of using AdaEraser for object removal across various scenarios, showcasing remarkable stability. For scenes where the object and background have highly similar textures, the adaptive suppression might inadvertently suppress background cues. We further provide examples involving two overlapping objects from the same class. Since the target object and the remaining content are highly similar in appearance and local texture, these cases are particularly challenging; nevertheless, AdaEraser still performs well.

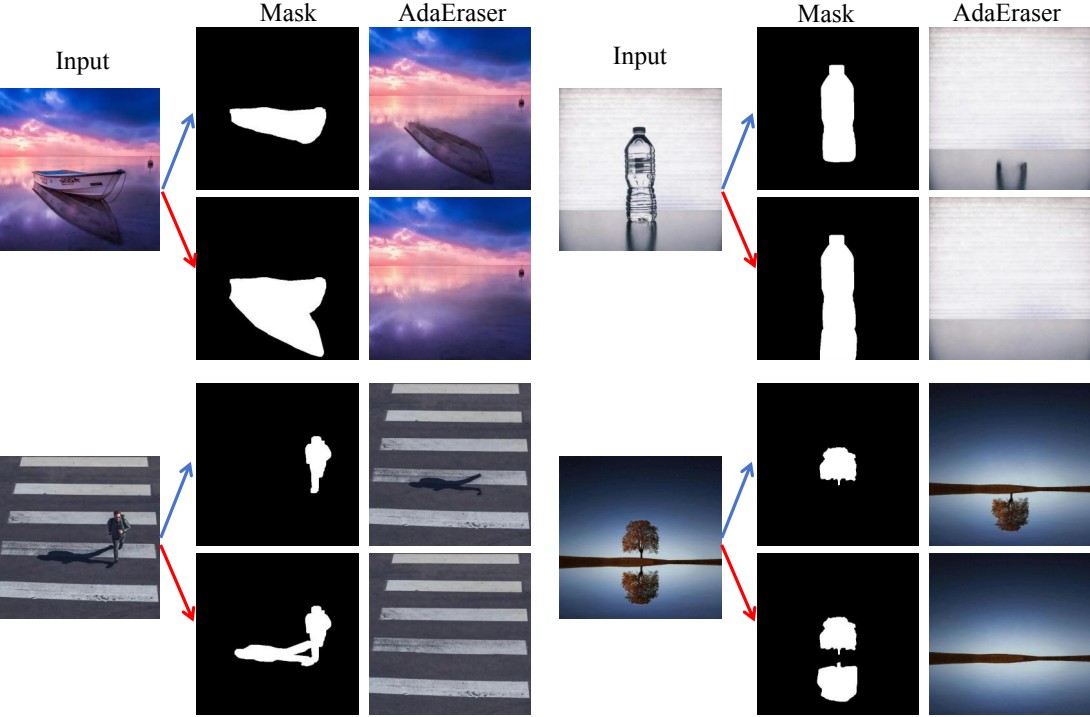

*Figure 11.* Object removal results under different masks. When the source image contains both the target object and its associated side effects, the removal quality depends heavily on the provided mask.

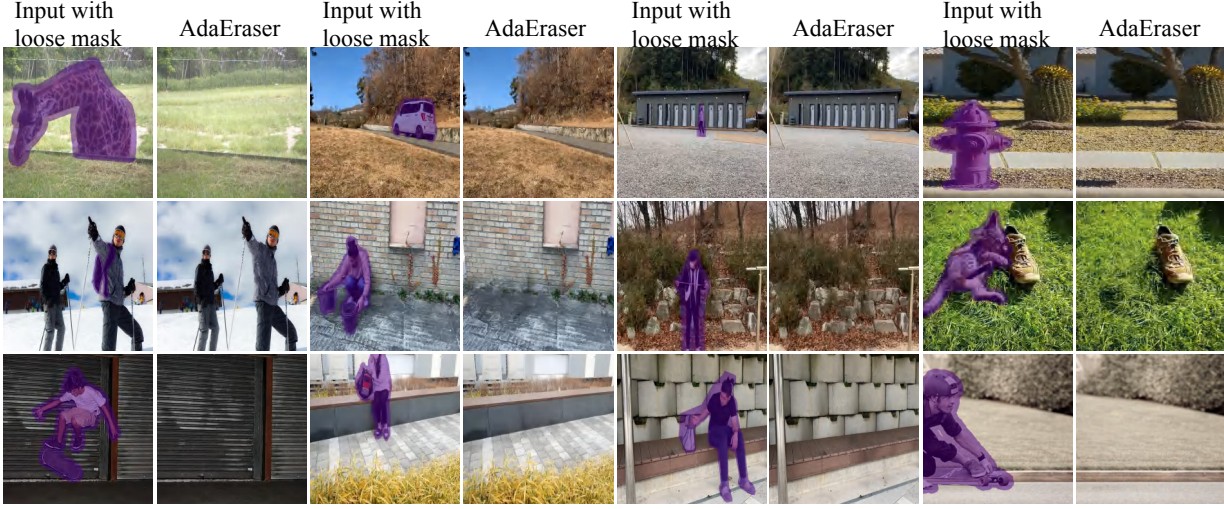

*Figure 12.* Performance under loose masks.

Input with Mask   w/ SD1.5   w/ SD2.1   w/ SDXL   w/ FLUX

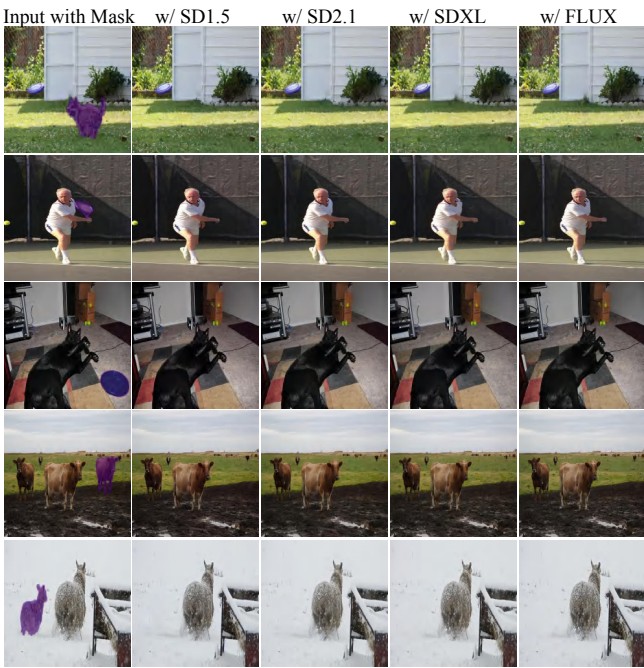

*Figure 13.* More results by our method, based on SD1.5, SD2.1, SDXL and FLUX.

## User study of object removal

The area covered by the light purple mask in the image indicates the object that is intended to be removed from the original image.

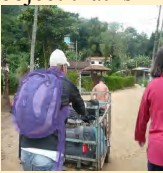

Please rank the options based on the overall quality of object removal, including the effectiveness of object removal, the restoration of the background in the removed region, and the preservation of the background in the untouched regions.

Rank each result on a scale from 1 to 8 (lower is worse).

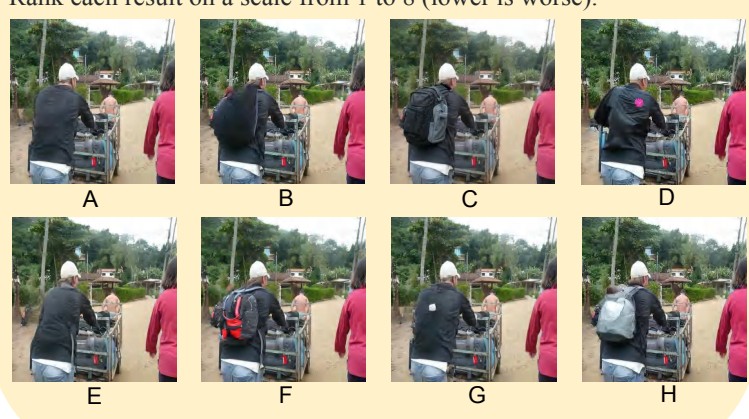

*Figure 14.* User study example. We provide an example, in which users are asked to rank the results of different models.

*High Overlap Between Object and Background*

*Complex-Scene Object Removal*

*Removal of Identical-Class Object*

*Multi-Object Removal*

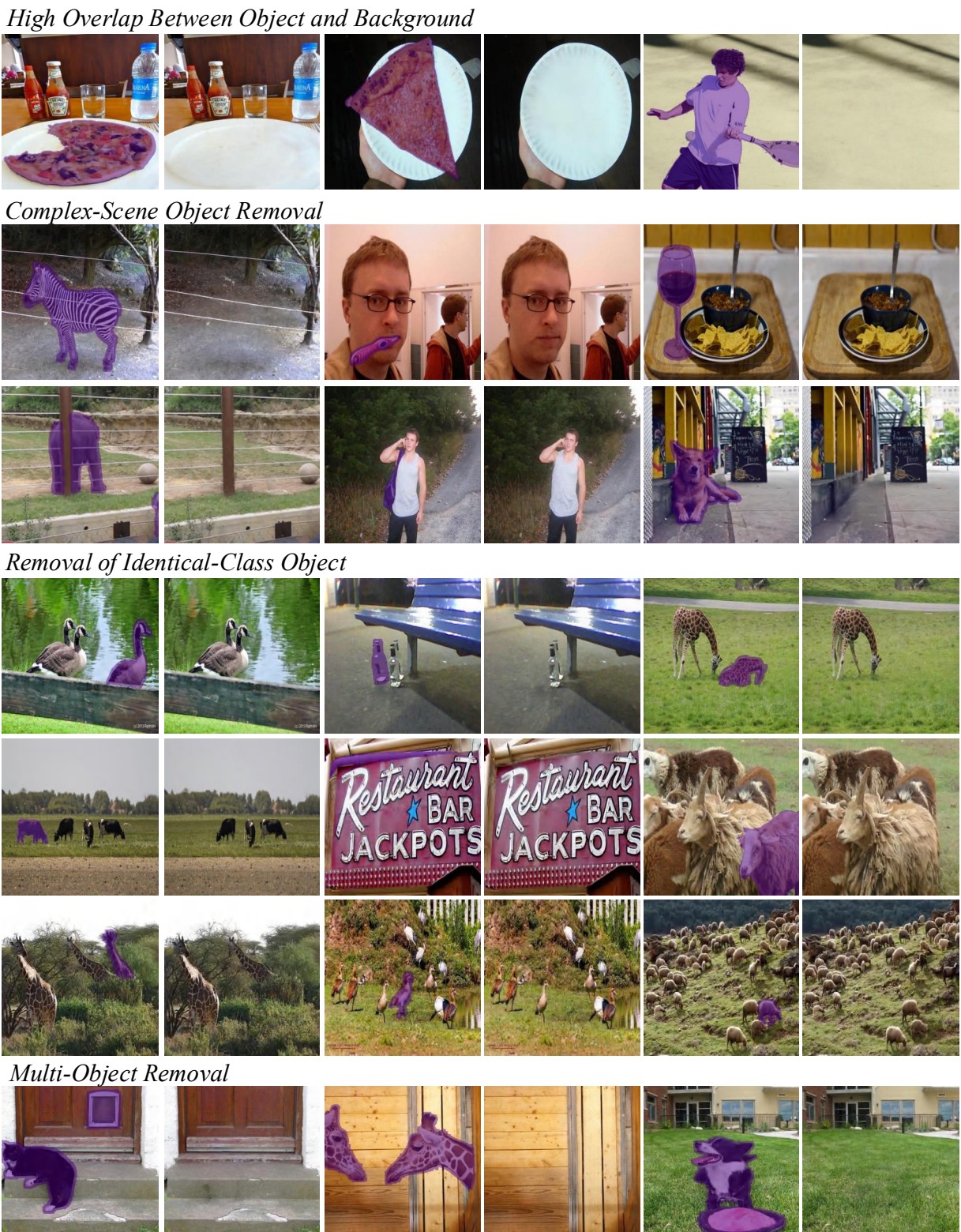

*Figure 15.* More results by AdaEraser.

