# OpenReview forum: "AdaEraser: Training-Free Object Removal via Adaptive Attention Suppression"
_ICML.cc/2026/Conference — ICML 2026 regular_

### Official Review · Reviewer_EMmT · 2026-03-02

**Soundness:** 2
**Presentation:** 3
**Significance:** 2
**Originality:** 2
**Overall Recommendation:** 4
**Confidence:** 4

**Summary:**

This paper proposed a training-free object removal method, AdaEraser, which mainly employs a token-wise adaptive attention suppression strategy. The experiments show that AadEraser achieves superior performance in object removal.

**Compliance With Llm Reviewing Policy:**

Affirmed.

**Final Justification:**

This paper presents a training‑free, mask‑based object removal model. Compared with mask‑free approaches, it requires dense masks, which can indeed be useful for interactive object removal. However, the current method performs poorly on distilled models, which limits its scalability and broader applicability.

The authors' rebuttal has addressed most of my concerns. I am raising my score to 4 (weakly accept).

**Key Questions For Authors:**

1. In the first row of Figure 1(c), the attentiveEraser method preserves the background more effectively (the grass remains clearer), whereas the grass in the AdaEraser result appears noticeably blurrier. What causes this difference?
2. Compared with current mainstream image editing methods (e.g., Qwen-image, Flux.2-Klein), what are the practical advantages of AdaEraser?
3. If the mask segmentation is inaccurate, mask-based methods may be negatively affected, whereas mask-free approaches could be more promising.

**Limitations:**

yes

**Strengths And Weaknesses:**

$\textbf{Strengths:}$
1. This paper proposed a training-free object removal framework that suppresses object-related regions based on attention maps.
2. The paper achieves strong performance on mask-based image object removal tasks.
3. The paper is clearly written and well presented.

$\textbf{Weakness:}$
1. The authors use SDXL as the base model and include comparison results with SD 1.5, SD 2.1, and FLUX in the appendix. The differences appear minimal. How does the proposed method perform on distilled accelerated models, such as 4‑step or even fewer‑step denoising models?

2. The mainstream image editing models today are predominantly mask‑free, such as Flux‑Kontext, Qwen‑Image‑Edit, and FLUX.2‑Klein, and therefore tend to have broader applicability. In contrast, mask‑based approaches are constrained by the accuracy of the masks. This limitation becomes more pronounced when the background contains complex textures (e.g., intricate clothing patterns) or when fine‑grained masks are required (e.g., character‑level text masks), where the current method may face notable restrictions.

---

> ### Author Rebuttal · Authors · 2026-03-28
>
> We sincerely appreciate your recognition of the strong performance and clear presentation of our work. Thank you very much for your valuable comments.
>
> We will address the concerns point by point.
>
> **Question 1:** Blur in Figure 1(c).
>
> **Answer:** Thank you for pointing this out. This difference is caused by an error in figure preparation. In Figure 1(c), the AdaEraser result in the first row was inadvertently saved through an intermediate interface, which introduced noticeable blur. The correct result is consistent with the image shown in **the bottom-right of Figure 4**, which corresponds to the same example and has normal sharpness. We sincerely apologize for this oversight and will correct the figure in the revised version.
>
> ---
>
> **Question 2:** Practical advantages of AdaEraser.
>
> **Answer:** Thank you for this important question. We agree that recent mask-free image editing models often have broader applicability and require less user interaction. At the same time, AdaEraser has a clear practical advantage in **precise instance-level removal**.
>
> Specifically, mask-free methods rely primarily on text instructions, and current image editing models may still struggle to accurately follow user intent under **complex spatial layouts or when multiple similar objects coexist**. In more extreme cases, text alone may be insufficient to uniquely specify one instance among many visually similar objects. In contrast, a mask provides **explicit spatial information**, making the editing target more precise and controllable.
>
> **Anonymous visual example:** https://anonymous.4open.science/r/ans-DD2C/comparison.png
>
> Therefore, while mask-free editing is more flexible in general scenarios, AdaEraser remains practically valuable when accurate localization and user-specified object removal are required.
>
> ---
>
> **Question 3 / Weakness 2:** Mask quality and mask-free alternatives.
>
> **Answer:** Thank you for your question. We agree that sensitivity to mask quality is a general limitation of **mask-based editing methods**, rather than a limitation unique to AdaEraser. Our method is designed within this setting because mask guidance provides explicit spatial controllability for object removal. Within this paradigm, we show initial robustness to **loose masks** in Appendix E, while **incomplete masks** remain more challenging, as discussed in Appendix D. We will clarify this distinction more explicitly in the revised version.
>
> We also agree that mask-free editing is an important and promising direction. Meanwhile, mask-based methods remain practically valuable when precise user-specified regional control is required. Exploring how to extend AdaEraser to better work with these mainstream editing models is an interesting direction for future work.
>
> ---
>
> **Weakness 1.1:** Cross-backbone generalization.
>
> **Answer:** Thank you for this comment. To further support cross-backbone generalization, we provide quantitative results on the Mulan dataset. The results show that AdaEraser generalizes well across SD1.5, SD2.1, FLUX, and SDXL, with consistently strong performance across metrics.
>
> | Architecture | FID ↓ | LPIPS ↓ | PSNR ↑ | ReMOVE ↑ | CFD ↓ |
> |---|---:|---:|---:|---:|---:|
> | SD1.5 | 55.981 | 0.2201 | 20.8360 | 0.9012 | 0.2057 |
> | SD2.1 | 54.905 | 0.2132 | 22.5313 | 0.9019 | 0.2043 |
> | FLUX | 52.335 | **0.2002** | 23.5690 | **0.9070** | 0.2008 |
> | SDXL | **51.108** | 0.2026 | **23.5871** | 0.9065 | **0.2002** |
>
> We use SDXL as the main model for its balance of quality and efficiency.
>
> ---
>
> **Weakness 1.2:** Performance on accelerated distilled models.
>
> **Answer:** Good point! To investigate this issue, we evaluate AdaEraser on SDXL-Lightning 4-step and provide quantitative results. While inference becomes significantly faster, the quantitative results show a clear degradation in removal and reconstruction quality compared with the standard SDXL setting. Qualitatively, we also observe noticeable artifacts within the edited regions.
>
> | Setting | LPIPS ↓ | PSNR ↑ | ReMOVE ↑ | CFD ↓ | Time (s) |
> |---|---:|---:|---:|---:|---:|
> | SDXL-Lightning (4-step) | 0.5055 | 15.1556 | 0.7083 | 0.2571 | 6.53 |
> | Standard SDXL | 0.2026 | 23.5871 | 0.9065 | 0.2002 | 15.41 |
>
> We believe this is because AdaEraser relies on sufficiently rich multi-step attention evolution during denoising. When the process is compressed to very few steps, the attention dynamics becomes less informative, leading to degraded restoration quality.
>
> Therefore, AdaEraser does not directly transfer well to accelerated distilled models. Improving compatibility with such models is an important direction for future work.

---

> > ### Author Rebuttal · Reviewer_EMmT · 2026-04-02
> >
> > Thank you for the authors’ rebuttal. I still have a few remaining questions.
> >
> > I conducted a simple test using the Flux.2-Klein-9B model to remove the giraffe on the far right (https://anonymous.4open.science/r/ans-DD2C/comparison.png). By modifying the prompt to “Only remove the giraffe on the most right, other giraffes remain,” the model is able to complete the removal. The random seed is 0, and the number of inference steps is 4. The test was performed on the demo website: https://huggingface.co/spaces/black-forest-labs/FLUX.2-klein-9B.
> >
> > Indeed, mask-based methods are necessary when fine-grained removal is required. However, I also tested selecting the third giraffe with a bounding box and using the prompt “Remove the giraffe selected in the red box, leaving all other giraffes unchanged.” The model is also able to complete the task. This suggests that instance-level removal may be achievable through simple prompts—rather than dense masks—by leveraging the model’s understanding capabilities.
> >
> > Regarding the weaker performance of AdaEraser on distilled models, this limitation may significantly restrict its practical applicability, since distilled variants are currently the most widely used versions of modern image models. Moreover, the mainstream trend in image modeling has shifted toward unified generation–editing models.

---

> > > ### Author Response · Authors · 2026-04-03
> > >
> > > We sincerely appreciate the reviewer’s thoughtful response and additional empirical observations. We would like to respectfully clarify that **mask-based object removal remains the mainstream setting in current object removal research** due to its precise instance-level controllability [1,2,3,4]. Our method is developed within this setting. The core contribution of our work is to analyze the evolution of self-attention maps in diffusion models during object removal, and based on this analysis, to design a method that is both **theoretically justified and empirically effective**. We will address your questions point by point.
> > >
> > > ---
> > >
> > > **Q1:** On the mask-based setting and comparison with instruction-based models.
> > >
> > > **A1:**  Thank you for your question. To address the reviewer’s new question more directly, we have added an additional comparison figure: https://anonymous.4open.science/r/compare_new-20DE/comparison_v2.png. For fairness and reproducibility, all tests involving Flux.2-Klein-9B were conducted on the demo website provided by the reviewer.
> > >
> > > Regarding the reviewer’s example, we tested Flux.2-Klein-9B with a random seed of 1895662641, using a semantically similar but slightly rephrased instruction: “Only remove the rightmost giraffe, other giraffes remain.” Although the meaning is nearly identical, the model still produces an incorrect removal result, as shown in Fig. (a). This suggests that Flux.2-Klein-9B remains **sensitive to prompt phrasing and may not be robust to small textual variations**; it is also sensitive to **the choice of random seed**. In practice, careful prompt tuning is often required to obtain the desired removal behavior. In contrast, AdaEraser provides more consistent removal behavior.
> > >
> > > We further provide more **representative examples**. As shown in Fig. (b), AdaEraser produces a plausible removal result, whereas Flux.2-Klein-9B with text-only guidance removes only part of the target, leading to semantically inconsistent output; under the reviewer’s bounding-box-plus-prompt setting, the model does not even successfully remove the object, which suggests that the model **may not reliably interpret the region indicated by the red box**.
> > >
> > > In Fig. (c), where multiple similar instances coexist, text alone cannot reliably specify the desired target for Flux.2-Klein-9B, while the bounding-box-plus-prompt setting also leads to incorrect regional removal. In contrast, AdaEraser yields a reasonable result. These examples suggest that **although instruction-based models are highly promising, current models still struggle to faithfully and consistently follow user intent in fine-grained object removal scenarios.** In contrast, mask-based methods remain more robust in such cases.
> > >
> > > ---
> > >
> > > **Q2:** Performance over distilled models and practical applicability.
> > >
> > > **A2:**  We also thank the reviewer for raising the important concern regarding performance on distilled models. AdaEraser is proposed as a training-free method for mask-based object removal, and within this setting it **achieves state-of-the-art performance while maintaining favorable efficiency (see Table 2), even outperforming some training-based method in both inference time and memory usage**. Therefore, we believe the current work still has clear practical value in the existing mask-based object removal setting.
> > >
> > > The key insight of AdaEraser is to adapt attention suppression according to the estimated object presence during denoising, which we believe provides a meaningful perspective for object removal. We agree that performance on distilled models is a meaningful limitation and extending AdaEraser to distilled models and unified generation-editing models is an important direction. We will explicitly clarify this as a limitation of the current work and a future direction in the revision.
> > >
> > > > [1] CLIPAway: Harmonizing Focused Embeddings for Removing Objects via Diffusion Models.
> > > > [2] SmartEraser: Remove Anything from Images using Masked-Region Guidance.
> > > > [3] OmniPaint: Mastering Object-Oriented Editing via Disentangled Insertion-Removal Inpainting.
> > > > [4] RORem: Training a Robust Object Remover with Human-in-the-Loop.

---

### Official Review · Reviewer_56ht · 2026-03-08

**Soundness:** 3
**Presentation:** 2
**Significance:** 3
**Originality:** 2
**Overall Recommendation:** 4
**Confidence:** 4

**Summary:**

The authors aim to explore a major problem in the field of image editing: achieving seamless object removal without the need for task-specific training or fine-tuning. The authors claim to examine an important concept termed "Adaptive Attention Suppression," which addresses the limitation of static masking in self-attention layers. The paper is technically sound and offers a practical, plug-and-play solution for current large-scale diffusion models.

**Compliance With Llm Reviewing Policy:**

Affirmed.

**Final Justification:**

After reading the response from the authors, I tend to keep my score rate at Weakly accept.

**Key Questions For Authors:**

- While it is training-free, the dual-path analysis or additional attention calculations might increase per-image inference time. A table comparing Latency/FPS with standard Inpainting methods would be beneficial.

- The performance likely depends on the quality of the initial user mask. Analysis of how "rough" vs. "precise" masks affect the results would strengthen the paper.

- For scenes where the object and background have highly similar textures, the adaptive suppression might inadvertently suppress background cues. More edge-case examples could be provided.

**Limitations:**

No, the authors should discuss the limitations of this work, making it easier for readers to understand the work deeply.

**Strengths And Weaknesses:**

Strengths:

- This work introduces a token-wise suppression strategy that dynamically balances object erasure and background reconstruction.
- A method that operates entirely during the inference phase is proposed, making it highly accessible for various pre-trained backbones (SD, FLUX).
- Experimental results verify the effectiveness of the proposed method.

Weaknesses:

- While it is training-free, the dual-path analysis or additional attention calculations might increase per-image inference time. A table comparing Latency/FPS with standard Inpainting methods would be beneficial.

- The performance likely depends on the quality of the initial user mask. Analysis of how "rough" vs. "precise" masks affect the results would strengthen the paper.

- For scenes where the object and background have highly similar textures, the adaptive suppression might inadvertently suppress background cues. More edge-case examples could be provided.

---

> ### Author Rebuttal · Authors · 2026-03-27
>
> We sincerely appreciate your positive feedback and constructive comments, especially the recognition of the value and effectiveness of our training-free framework.
>
> Below, we address the concerns point by point.
>
> **Question 1:** A table comparing Latency/FPS with standard Inpainting methods would be beneficial.
>
> **Answer:** Thank you for this suggestion. We agree that efficiency is important. We  report an efficiency comparison in the main paper (**Lines 423--428, Table 2**). We report the **average inference time per image**, which directly reflects the latency concern raised by the reviewer. AdaEraser takes **15.41s** per image on average, compared with **13.98s** for AttentiveEraser and **20.37s** for OmniPaint, while keeping memory overhead within **15%** relative to AttentiveEraser.
>
> We will make this comparison more explicit in the revised version.
>
> ---
>
> **Question 2:** The performance likely depends on the quality of the initial user mask. Analysis of how "rough" vs. "precise" masks affect the results would strengthen the paper.
>
> **Answer:** Thank you for your feedback. We already provide qualitative analysis under **loose masks** in **Appendix E**, where AdaEraser remains stable even when the mask extends beyond the target object. To further support this point, we additionally evaluate a loose-mask setting on OABench by dilating the object masks. AdaEraser still achieves the best performance, outperforming AttentiveEraser and RORem.
>
> | Methods | FID ↓ | LPIPS ↓ | PSNR ↑ | ReMOVE ↑ |
> |---|---:|---:|---:|---:|
> | AttentiveEraser | 40.771 | 0.1711 | 22.9818 | 0.8147 |
> | RORem | 40.532 | 0.1615 | 23.0197 | 0.8171 |
> | AdaEraser | **39.829** | **0.1602** | **23.1656** | **0.8215** |
>
> This is consistent with our design: AdaEraser progressively decays suppression on masked tokens while preserving global attention, which better aligns with the generative prior of diffusion models. Therefore, even when loose masks include some background tokens, the model can still leverage full-image context for recovery.
>
> In contrast, **incomplete masks** are more challenging, since the **foreground-background blending** depends on the mask to **define the edited region**. If the mask fails to fully cover the object or its side effects (e.g., shadows or reflections), the removal quality may degrade, as discussed in **Appendix D**.
>
> Overall, AdaEraser remains effective when the mask is accurate or slightly loose, but is more sensitive when the mask is incomplete. We will make the effect of mask quality more explicit in the revised version, in particular by clarifying how “rough” masks may correspond to loose versus incomplete masks, which affect AdaEraser differently.
>
> ---
>
> **Question 3:** For scenes where the object and background have highly similar textures, the adaptive suppression might inadvertently suppress background cues. More edge-case examples could be provided.
>
> **Answer:** We greatly appreciate your valuable suggestion. We agree that scenes where the target object and background share highly similar textures are potentially challenging. We have discussed **several related challenging cases** in our analysis. In particular, **Appendix C** shows that reconstruction quality may degrade when the background to be recovered contains complex structure or texture. In addition, **Figure 15** includes several visually challenging cases with strong object-background interaction, where AdaEraser still performs well in many examples.
>
> We additionally provide two examples with **two overlapping same-class objects**, where the front one is specified for removal. Since the target object and the remaining content are highly similar in **appearance and local texture**, these are challenging cases, yet AdaEraser still performs well. **Anonymous visual example:** https://anonymous.4open.science/r/case-CC71/example.png
>
> We will clarify in the revised version that highly similar object-background textures are a challenging setting for the proposed proxy, and include more discussion of such edge cases.
>
> ---
>
> **Discussion of limitations**
>
> **Answer:** Thank you for your valuable feedback. We agree that the limitations should be discussed more explicitly. We analyze related limitations in the appendix: **Appendix C** shows failure cases when the background to be recovered contains **complex structure or texture**, and **Appendix D** discusses degradation caused by **incomplete masks**.
>
> These observations are consistent with the reviewer’s concerns. We will make these limitations more explicit in the revised version.

---

> > ### Author Rebuttal · Reviewer_56ht · 2026-04-02
> >
> > Thanks to the authors for the detailed response. My concerns are fully solved. I hope the author will incorporate the above results into the next version of the paper.

---

> > > ### Author Response · Authors · 2026-04-02
> > >
> > > Thank you very much for your encouraging feedback. We are glad that our rebuttal has addressed your concerns.
> > >
> > > We also appreciate your constructive suggestions, and we will incorporate the discussed results and corresponding clarifications into the next version of the paper.

---

### Official Review · Reviewer_EC2f · 2026-03-10

**Soundness:** 3
**Presentation:** 4
**Significance:** 3
**Originality:** 3
**Overall Recommendation:** 5
**Confidence:** 3

**Summary:**

This paper proposes a training-free object removal method, AdaEraser, that adaptively suppresses self-attention during diffusion-based removal. The key idea is that fixed suppression may help erase the target object but can also harm background reconstruction. To address this, the authors estimate a token-wise object presence score from the similarity between source and denoising self-attention maps, and use it to dynamically control suppression strength. Experiments show strong results against both training-free and training-based baselines.

**Compliance With Llm Reviewing Policy:**

Affirmed.

**Final Justification:**

This paper is really interesting, and the authors have resolved my concerns, so I raised my score from 4 to 5.

**Key Questions For Authors:**

1. The method relies on token-wise self-attention similarity as a proxy for object presence. Could the authors discuss more explicitly under what conditions this proxy may fail, for example, in scenes with similar nearby objects, repeated textures, or complex background reconstruction?
2. The paper shows that attention-map similarity correlates with removal progress, but this still seems largely empirical. Could the authors clarify whether they view the presence score as a heuristic control signal or as a more principled estimator of object presence?

**Limitations:**

Yes.

**Strengths And Weaknesses:**

Strengths:
The paper is well motivated and focuses on a concrete technical issue. The central idea is clear, and the observation is interesting. The method is relatively simple, does not require additional training, and is easy to understand once the presence-score idea is introduced. The empirical results are strong, and the analysis experiments are mostly aligned with the proposed mechanism.

Weaknesses:
The proposed presence score remains an empirical proxy rather than a rigorously validated measure of object presence. Changes in self-attention similarity may reflect not only whether the object is still present, but also denoising dynamics, background reconstruction, or interactions with similar structures. So while the idea is intuitive and seems effective, its interpretation may be somewhat stronger than what is strictly established.

---

> ### Author Rebuttal · Authors · 2026-03-27
>
> We sincerely **appreciate your positive assessment of our work, especially your recognition of its motivation, clarity, and empirical strength**. Thank you very much for your thoughtful and constructive comments!
>
> Below, we will address all the concerns point by point.
>
> **Question 1:** The method relies on token-wise self-attention similarity as a proxy for object presence. Could the authors discuss more explicitly under what conditions this proxy may fail, for example, in scenes with similar nearby objects, repeated textures, or complex background reconstruction?
>
> **Answer:** Thank you for raising this important question. We agree that token-wise self-attention similarity has limitations as a proxy for object presence, and we also observe that its performance degrades when **the background behind the removed object contains complex structure**. In this case, the challenge lies not only in estimating whether the object is still present, but also in recovering a background that is structurally correct and consistent with the surrounding context.
>
> As discussed in Appendix A, our adaptive suppression mainly regulates **the suppression strength** within the masked region to balance object removal and background restoration, rather than explicitly determining the exact background structure to be recovered. Therefore, when the occluded background itself involves complex structure, even a reasonable suppression signal may still fail to guide the model toward a correct structural reconstruction, resulting in erroneous background generation, structural distortions, or missing details. This is also consistent with the failure cases shown in **Appendix C**. We will clarify this limitation more explicitly in the revision.
>
> We also agree that scenes with **similar nearby objects and texture-rich backgrounds** can be challenging, and we have considered such cases in our analysis. As shown in **Figure 15**, AdaEraser still performs well in many related challenging scenarios. We will discuss these cases more explicitly in the revised version.
>
> ---
>
> **Question 2:** The paper shows that attention-map similarity correlates with removal progress, but this still seems largely empirical. Could the authors clarify whether they view the presence score as a heuristic control signal or as a more principled estimator of object presence?
>
> **Answer:**  Thank you for your question. We **do not** view the presence score as **a strict estimator of true object presence**. In fact, the degree of object existence is not directly quantifiable in the noisy latent code. Therefore, our method should be understood more as **a heuristic, control-oriented proxy**: it provides a stable and useful signal for regulating suppression strength during denoising.
>
> Meanwhile, it is not a purely ad hoc heuristic. Appendix A provides **theoretical motivation** for why this signal is meaningful and stable in our setting. In particular, Appendix A.1 explains why the cosine similarity of token-wise self-attention maps can serve as an approximate semantic indicator, Appendix A.2 discusses its robustness under diffusion noise, and Appendix A.3 provides an interpretive view of adaptive suppression as principled attention reweighting. Our experiments further validate the effectiveness of the proposed mechanism.
>
> In conclusion, the presence score is a **theoretically justified and empirically effective** control signal, rather than a strict estimator of true object presence.
>
> ---
>
>
> **Weakness:**  While the idea is intuitive and seems effective, its interpretation may be somewhat stronger than what is strictly established.
>
> **Answer:**  Thank you for your valuable feedback. We agree that the proposed presence score is better understood as **a heuristic, control-oriented proxy** for object presence, rather than as a rigorously validated or directly measurable quantity in the noisy latent space. It is not intended as a strict estimator, but as a practical signal for adaptive suppression. **Appendix A** provides **theoretical motivation** for why this signal is meaningful in our setting, while **Figures 3 and 7** and **Tables 3 and 4** provide empirical and functional support.
>
> We agree that this point should be stated more carefully in the paper, and we **appreciate the reviewer’s reminder**. We will clarify this positioning more explicitly in the revised version.

---

> > ### Author Rebuttal · Reviewer_EC2f · 2026-04-01
> >
> > This paper is really interesting, and the authors have resolved my concerns.

---

> > > ### Author Response · Authors · 2026-04-01
> > >
> > > Thank you for your prompt response and positive feedback on our revisions. Please let us know if you have any further questions or suggestions.
> > >
> > > We also appreciate your valuable contributions throughout the review process!

---

### Official Review · Reviewer_P1qq · 2026-03-13

**Soundness:** 3
**Presentation:** 3
**Significance:** 3
**Originality:** 2
**Overall Recommendation:** 4
**Confidence:** 4

**Summary:**

AdaEraser, a training-free object removal method for diffusion models. The method builds on attention suppression techniques but argues that fixed suppression across the denoising process harms background reconstruction. The key idea is to adaptively adjust suppression strength using token-wise presence estimation, computed by comparing self-attention maps between a reference branch and the current denoising branch. When the target object fades during denoising, suppression is relaxed to allow the model to synthesize background details more naturally. Experiments on Mulan and OABench show improvements over both training-free and training-based baselines.

**Compliance With Llm Reviewing Policy:**

Affirmed.

**Key Questions For Authors:**

How reliable is the cosine similarity between attention maps as a proxy for object presence across different object categories, scene layouts, and diffusion timesteps? Is there direct quantitative evidence validating this assumption?
The improvements over strong baselines such as RORem are relatively small on several metrics. Are these gains statistically significant, and how stable are the results across multiple runs or random seeds?
The experiments focus on two benchmarks with ground-truth masks. How does the method perform in more challenging scenarios, such as large removal regions, cluttered scenes, or thin object structures?

**Limitations:**

The method depends on accurate object masks, and removal quality can degrade when the mask does not cover object side effects such as shadows, reflections, or surrounding context. The proposed object presence estimation relies on cosine similarity between attention maps, but this assumption may not always hold in scenes with complex textures or overlapping objects where attention patterns are ambiguous. The evaluation is limited to two datasets (Mulan and OABench) and does not include systematic analysis for challenging cases such as large masks, thin structures, multiple objects, or highly cluttered scenes.

**Strengths And Weaknesses:**

Treating object removal as a gradual process during denoising is a useful perspective and addresses a real limitation of earlier suppression-based approaches. The method is also computationally lightweight since it operates directly on attention maps without additional networks or retraining. This makes it appealing for practical deployment.

The method mainly extends prior attention-suppression approaches (e.g., AttentiveEraser) by introducing an adaptive suppression weight, while the overall architecture and pipeline remain largely unchanged. The object presence estimation mechanism is insufficiently validated, since the paper assumes that cosine similarity between attention maps reliably indicates object presence but does not provide direct quantitative evidence for this assumption. The performance gains over strong baselines (e.g., RORem) are relatively small on several metrics, making it unclear whether the improvements are statistically significant. The evaluation scope is narrow, as experiments are limited to two benchmarks and do not systematically analyze challenging cases such as large masks, cluttered scenes, or thin object structures. The robustness to imperfect masks is not thoroughly studied, despite the method depending on accurate masks for effective removal. Finally, the cross-model generalization claim is weakly supported, as results on other diffusion backbones are only qualitative and lack quantitative comparison.

---

> ### Author Rebuttal · Authors · 2026-03-27
>
> We sincerely appreciate your recognition of the key motivation and practical value of AdaEraser, as well as your thoughtful and constructive feedback. We address concerns point by point.
>
> **Question 1 / Weakness 2 :** Validation of the object presence estimation mechanism.
>
> **Answer:** Thank you for your question. Direct validation against a latent-space “ground-truth object existence” is difficult because such a quantity is not directly measurable. The presence score is not intended as an exact measure of object existence in the noisy latent space, but as a theoretically justified and empirically validated approximate proxy for adaptive self-attention suppression.
>
> We support this proxy in three ways: **Appendix A.1--A.2** provides theoretical justification; **Figures 3 and 7** show consistent decreasing trends across examples, tokens, layers, and timesteps as removal progresses; **Tables 3 and 4** show that replacing this signal with simpler alternatives degrades performance.
>
> ---
>
> **Question 2 / Weakness 3:** Statistical significance and stability of quantitative comparisons.
>
> **Answer:** Good question! To verify robustness, we additionally ran the evaluation with **5 seeds** on **Mulan**  and observed consistently better performance with small variance across runs.
>
> | Method | LPIPS ↓ | PSNR ↑ | ReMOVE ↑ | CFD ↓ |
> |---|---:|---:|---:|---:|
> | AttentiveEraser | 0.2228 ± 0.0013 | 22.5936 ± 0.1049 | 0.8998 ± 0.0005 | 0.2097 ± 0.0013 |
> | RORem | 0.2087 ± 0.0008 | 23.4878 ± 0.0551 | 0.9046 ± 0.0003 | 0.2037 ± 0.0011 |
> | OmniPaint | 0.2317 ± 0.0057 | 21.4023 ± 0.1271 | 0.8703 ± 0.0005 | 0.2324 ± 0.0015 |
> | AdaEraser | **0.2027 ± 0.0011** | **23.5647 ± 0.0431** | **0.9064 ± 0.0002** | **0.2001 ± 0.0012** |
>
> To further assess significance, we performed **paired t-tests** and **Wilcoxon signed-rank tests** against the above baselines on per-image **PSNR** and **ReMOVE**. The improvements are statistically significant under both tests (**all p < 0.01**).
>
> ---
>
> **Question 3 / Weakness 4 / Limitation 2 / Limitation 3:** Evaluation scope and challenging scenarios.
>
> **Answer:** **Mulan** and **OABench** are not limited in scope: their construction already covers diverse object categories, scene layouts, and complex real-world cases.
>
> We agree that complex scenes are more challenging, and we provided related experiments and analysis. **Figures 6 and 15** cover several representative difficult cases, including large removal regions, thin object structures, high-overlap scenes, and visually complex scenes, where AdaEraser still performs well in many examples. We also provide failure-case analysis in **Appendix C**, showing AdaEraser may degrade when the region to be recovered contains complex texture or structure. We will discuss these scenarios more explicitly in the revised version.
>
> ---
>
> **Weakness 1:** Limited architectural novelty.
>
> **Answer:** We agree that AdaEraser follows the overall framework of prior attention-suppression-based object removal methods. Our contribution is therefore not a new architecture, but a new control mechanism within this paradigm.
>
> The key difference is that suppression is no longer fixed throughout denoising. Instead, we introduce a parallel unsuppressed reference branch and use reference--target attention comparison to guide token-wise adaptive suppression, which better balances object removal and background reconstruction. Although lightweight, this change leads to meaningful improvements.
>
> ---
>
> **Weakness 5:** Robustness to imperfect masks.
>
> **Answer:** As shown in **Appendix E**, we evaluate AdaEraser under loose masks generated by morphological dilation. The results show that AdaEraser remains reasonably stable even when the mask extends beyond the object boundary. We additionally evaluate a **loose-mask setting on OABench** by dilating the masks. AdaEraser still achieves the best performance.
>
> | Method | FID ↓ | LPIPS ↓ | PSNR ↑ | ReMOVE ↑ |
> |---|---:|---:|---:|---:|
> | AttentiveEraser | 40.771 | 0.1711 | 22.9818 | 0.8147 |
> | RORem | 40.532 | 0.1615 | 23.0197 | 0.8171 |
> | AdaEraser | **39.829** | **0.1602** | **23.1656** | **0.8215** |
>
> At the same time, due to the foreground--background blending design, removal quality may degrade when the mask is incomplete and fails to cover side effects, as discussed in **Appendix D**. We will clarify this scope and limitation more explicitly in the revised paper.
>
> ---
>
> **Weakness 6:** Quantitative cross-model generalization.
>
> **Answer:** Thank you for your comment. We provide quantitative results on the Mulan dataset, which show promising generalization across different diffusion backbones. We use SDXL as the main model for its balance of quality and efficiency.
>
> | Architecture | FID ↓ | LPIPS ↓ | PSNR ↑ | ReMOVE ↑ |
> |---|---:|---:|---:|---:|
> | SD1.5 | 55.981 | 0.2201 | 20.8360 | 0.9012 |
> | SD2.1 | 54.905 | 0.2132 | 22.5313 | 0.9019 |
> | FLUX | 52.335 | **0.2002** | 23.5690 | **0.9070** |
> | SDXL | **51.108** | 0.2026 | **23.5871** | 0.9065 |

---

### Decision · Program_Chairs · 2026-04-30

**Decision:**

Accept (regular)

**Comment:**

In this paper, the authors presented a training-free adaptive framework for object removal. The paper was reviewed by four expert reviewers, followed by a rebuttal, an author-reviewer discussion phase, and an AC-reviewer discussion phase. The paper received an overall positive rating with 1 Accept and 3 Weak Accept.

The reviewers agree on the motivation, the simple and effective idea, the strong performance and the overall quality of writing. Good contributions could be made to the community, as appreciated by the reviewers.
There were some concerns regarding the weaknesses of the paper, including the scalability of the proposed method, insufficient validation and analysis, a lack of supporting evidence for some claims, the efficiency and some other minor issues. After the rebuttal and the author-reviewer discussions, most concerns were well addressed, as acknowledged by the reviewers. The AC-reviewer discussions further confirm the strengths of this paper.
Overall, the paper is technically sound, well-written, and could contribute to and be of interest to at least some fraction of the ICML community.

As a result, the AC is happy to recommend acceptance of this paper, while asking the authors to incorporate the revisions and the necessary additional justifications in the rebuttal/discussions to their final version.